# Aerosol absorption retrieval at ultraviolet wavelengths in a

- 2 complex environment
- 3
- 4 Stelios Kazadzis<sup>1,2</sup>, Panagiotis I. Raptis<sup>2</sup>, Natalia Kouremeti<sup>1</sup>, Vassilis Amiridis<sup>3</sup>,
- 5 Antti Arola<sup>4</sup>, Evangelos Gerasopoulos<sup>2</sup>, Gregory L. Schuster<sup>5</sup>
- 6 [1]{Physikalisch-Meteorologisches Observatorium Davos, World Radiation Center
- 7 (PMOD/WRC) Dorfstrasse 33, CH-7260 Davos Dorf, Switzerland}
- 8 [2] {Institute of Environmental Research and Sustainable Development, National Observatory
  9 of Athens, Greece}
- 10 [3] {Institute of Astronomy Astrophysics, Space Applications and Remote Sensing, National
- 11 Observatory of Athens, Greece}
- 12 [4] {Finnish Meteorological Institute, Kuopio Unit, Finland}
- 13 [5] {NASA Langley Research Center, Hampton, VA, USA}
- 14 Correspondence to: stelios.kazadzis@pmodwrc.ch
- 15

# 16 Abstract

17 We have used total and diffuse UV irradiance measurements with a multi-filter rotating 18 shadow-band radiometer (UVMFR), in order to calculate aerosol absorption properties 19 (Single Scattering Albedo - SSA) in the UV range, for a 5 years period in Athens, Greece. 20 This data set was used as input to a radiative transfer model and the SSA for 368nm and 21 332nm has been calculated. Retrievals from a collocated CIMEL sun-photometer were used 22 to validate the products and study absorption spectral behavior SSA values at these 23 wavelengths. UVMFR SSA together with synchronous, CIMEL-derived, retrievals at 440nm, 24 show a mean of 0.90, 0.87 and 0.83, with lowest values (higher absorption) towards lower 25 wavelengths. In addition, noticeable diurnal variations of the SSA in all wavelengths are 26 revealed, with amplitudes in up to 0.05. High SSA wavelength dependence is found for cases 27 of low Ångström exponents and also an SSA decrease with decreasing extinction optical

- 1 depth, suggesting an effect of the different aerosol composition. Dust and Brown Carbon UV
- 2 absorbing properties were investigated to understand seasonal variability of the results.
- 3

## 4 1 Introduction

5 The role of aerosols, both natural and anthropogenic, is extremely important for regional and 6 global climate change studies as well as for overall pollution mitigation strategies (e.g 7 IPCC,2013). However, a considerable amount of work still needs to be carried out, 8 particularly as it appears that climate change is accelerating with aerosols impacting at local, 9 regional and global scales. Furthermore, the components controlling aerosol forcing, account 10 for the largest uncertainties in relation to anthropogenic climate change (IPCC, 2007, IPCC, 11 2014). A comprehensive review of the assessment of the aerosol direct effect, its state of play 12 as well as outstanding issues, is given by (IPCC, 2014) and (Yu et al., 2006). Both emphasize that the significant aerosol absorption uncertainties in global Single Scattering Albedo (SSA), 13 14 constitute one of the largest single source of uncertainty in current modeling estimates of 15 aerosol climate forcing. SSA is the ratio of scattering to total extinction (scattering plus 16 absorption), and it depends strongly on chemical composition, particle size, mixture, relative 17 humidity and wavelength. Comprehensive measurements are crucial to understand their 18 effects and to reduce SSA uncertainties that propagate into aerosol radiative forcing estimates. 19 For example for the same aerosol load (aerosol optical depth), the absorbing nature of 20 aerosols can lead to up to 50% change in the erythermal irradiance, compared to only scattering aerosols (Bais et al, 2014). SSA calculated here differs from in situ SSA values 21 22 retrieved from absorption and scattering measurements at a single altitude level (e.g. at the 23 ground), in that it is a columnar measurement, arising from solar irradiance attenuation along 24 a fixed irradiance path.

25 In the visible (VIS) part of the spectrum, advanced retrieval algorithms for microphysical 26 aerosol properties have been developed in the framework of the Aerosol Robotic Network 27 (AERONET) and the Skyradiometer Network (SKYNET) (e.g., Dubovik and King, 2000; 28 Nakajima et al., 1996) All AERONET stations currently provide inversion based VIS-SSA 29 retrievals. In addition, Goering et al. (2005), Taylor et al (2008) and Kudo et al. (2008) have 30 proposed estimation techniques for the retrieval of spectral aerosol optical properties by 31 combining multi-wavelength measurements using a priori constraints that are applied 32 differently than in the single wavelength methods. SSA retrieval in the ultraviolet (UV) part

1 of the spectrum is weaker with large uncertainties. As AERONET does not provide any 2 information about SSA at the UV, compared to the visible spectral region, only a few 3 publications have dealt with aerosol absorption at UV wavelengths (e.g. Eck et al., 1998; Krotkov et al., 2005a; Bais et al., 2005; Corr et al., 2009). It is envisaged that improvement in 4 5 measurement precision and in the general understanding of aerosol absorption in the UV (and 6 immediate derivatives like the SSA) in various scientific applications, will contribute 7 significantly to enhancing the accuracy of radiation forcing estimates. For example, desert 8 dust particles (Alfaro et al., 2004), soot produced by fossil fuel burning, and urban 9 transportation, all strongly absorb UV radiation. However, optical properties of other potential 10 UV absorbers like organic, nitrate and aromatic aerosols are still poorly known. Bergstrom et 11 al., 2003 showed that spectra of aerosol SSA obtained in different campaigns around the 12 world differed significantly from region to region, but in ways that could be ascribed to regional aerosol composition. Moreover, results from diverse air, ground, and laboratory 13 14 studies, using both radiometric and in situ techniques, show that the fractions of black carbon, 15 organic matter, and mineral dust in atmospheric aerosols play a role in the determination of 16 the wavelength dependence of aerosol absorption (Russell et al., 2010). Barnard et al. (2008), 17 investigating the variability of SSA in a case study for the Mexico City metropolitan area, 18 found that, in the near-UV spectral range (250 to 400 nm), SSA is much lower compared to 19 SSA at 500 nm indicative of enhanced absorption in the near-UV range. They suggested that 20 absorption by elemental carbon, dust or gas alone could not account for this enhanced 21 absorption leaving the organic carbon component of the aerosol as the most likely absorber. 22 It has been found in many studies that, in addition to dust, the absorbing organic carbon 23 compounds can induce strong spectral absorption increasing towards the shortest UV 24 wavelengths. Sources of these light-absorbing organic carbon compounds (often called as 25 "Brown Carbon", BrC) are various; biomass burning (e.g. Kirchstetter et al.2004), urban 26 smoke (e.g. Liu et al. 2015) and biogenic emissions (e.g. Flores et al. 2014).

Moosmuller et al (2012) showed that iron concentration in mineral dust aerosols is linked to lower SSA at 405nm than in 870, which could be a hint for lowest SSA in the UV-VIS range during dust events. Medina et al (2012) found in El Paso-Juarez also large variation in UV range SSA, with lower values than visible wavelengths and showed that on heavy polluted days it can get as low as 0.53 at 368nm. An effort was made to calculate SSA in lower UV wavelengths, using Brewer measurements, at Belgium, revealing lowest values but with high uncertainty (Nikitidou et al, 2013). Recently Schuster et al (2016) have tried to distinguish

aerosol types, by their optical properties and assumed that dust particles have higher
 absorption at UV wavelengths, and used imaginary refractive index spectral dependence to
 separate from black carbon and infer hematite/goethite in the coarse mode. They found that
 dust particles containing hematite are highly absorbing in the UV region.

5 Ultraviolet (UV) solar radiation has a broad range of effects on life on Earth (UNEP et al., 1998;UNEP et al., 2007;UNEP, 2003). It influences not only human beings (e.g. (Diffey, 6 7 1991)), but also plants and animals (e.g. Bornman and Teramura, 1993). Furthermore, it 8 causes degradation of materials and functions as a driver of atmospheric chemistry. There are 9 various studies linking changes of the UV radiation field with changes in the scattering and 10 absorption of aerosols in the atmosphere (e.g. Zerefos et al., 2012). Such changes can be 11 comparable in magnitude with those caused by the decline in stratospheric ozone (Elminir, 12 2007; Reuder and Schwander, 1999; Krotkov et al., 1998). As an example, analysis of long 13 term UV time series at Thesaloniki, Greece, showed a reduction of 7% of AOD per decade 14 was recorded, but the UV Irradiance has increased by 9% (after removing ozone column 15 effect on it) which could only be explained by change in the absorption characteristics of 16 aerosols in the area (Meleti et al, 2009). Moreover, UV variations caused by changes in 17 aerosol optical properties directly affect tropospheric photochemistry:

- increases in regional O<sub>3</sub> (10-20 ppb for Eastern USA) caused by increased UV levels
   due to the presence of non-absorbing aerosols (Dickerson et al., 1997).
- decreases in regional O<sub>3</sub> (up to 50 ppb for Mexico City and for particular days) caused
   by strong UV reduction due to absorbing aerosols (Castro et al., 2001).
- There are also several more scientific issues that may be clarified with accurate knowledge ofaerosol absorption properties:

Aerosol effects on UV trends may enhance, reduce or reverse effects of stratospheric
 ozone change

Future scenarios for simulations of global UV levels are based on ozone recovery, having as their sole input the predicted future decline in columnar ozone. Furthermore, simulations of observed tendency of reduced anthropogenic aerosols in the atmosphere in the US and Europe during the course of the last decade (den Outer et al., 2005) included only cloud and AOD changes in the characterization of likely UV trends. In this regard, changes in the absorbing properties of aerosols on global scales would have had a large effect on the uncertainty budget

in any of the above simulations (WMO, 2003). For example, a decrease in aerosol absorption
 properties accompanied by an AOD decrease in Europe could lead to a significant
 acceleration of the calculated ozone decline related to UV upward trends (Kazadzis et al.,
 2009, Zerefos et al., 2012).

Solar irradiance satellite retrieval algorithms are directly affected by the presence of
absorbing aerosols

7 The discrepancies between ground-based (GB) UV measurements and satellite-derived (OMI, 8 TOMS, GOME) data are directly related to aerosol absorption that is absent from satellite 9 retrieval algorithms (Tanskanen et al., 2007;Arola et al., 2005). It has been shown that 10 enhanced aerosol UV absorption in urban areas can cause up to 30% overestimation in the 11 satellite retrieved UV radiation (Kazadzis et al., 2009).

12 • Uncertainty on commonly used atmospheric radiative transfer applications and codes

Radiative transfer algorithms calculating UV irradiance, fall short in precision due to large 13 14 uncertainties in the input parameters (e.g. levels of ozone, aerosol composition and the surface 15 albedo) used in model calculations. It is now known that the major input source of uncertainty 16 in radiative-transfer model simulations, is aerosol absorption (e.g. Van Weele et al., 2000). In particular, the direct radiative effect of aerosols is very sensitive to SSA. For example, a 17 18 change in SSA from 0.9 to 0.8 can often alter the sign of the direct effect (Yu et al., 2006). 19 Furthermore, availability and quality of observational SSA data do not match with those available for AOD (Krotkov et al., 2005a). This is compounded by the lack of information on 20 21 the vertical profile of aerosol optical properties such as the SSA at global scales. Only few 22 case studies have dealt with such measurements and have been limited to local scales (Müller 23 et al., 1999).

a. The major parameters that describe radiation and aerosol interactions are the aerosol optical depth (τ), the SSA and the asymmetry parameter (g). The aerosol optical depth at a
wavelength λ is the integral of the aerosol extinction coefficient (b<sub>ext</sub>(λ)) over a certain atmospheric layer (in the height range z<sub>1</sub> to z<sub>2</sub>).

28 b.

29 
$$\tau = \int_{z_1}^{z_2} b_{ext}(\lambda) \cdot dz \tag{1}$$

5

1 The SSA at a wavelength  $\lambda$  provides the contribution of aerosol particle scattering relative to 2 the total extinction (absorption plus scattering),

3 
$$SSA = \frac{b_{sca}(\lambda)}{b_{abs}(\lambda) + b_{sca}(\lambda)}$$
(2)

4 Values for the SSA range from 0 (absorbing aerosols only) to 1 (no absorption). The 5 asymmetry parameter, is the phase function (P) weighted average of the cosine of the 6 scattering angle ( $\theta$ ) over all directions. Assuming azimuthal symmetry, the scattering angle 7 integration extends from –  $\pi$  to + $\pi$  such that the asymmetry parameter (g) is given by

8 
$$g = \frac{1}{2} \cdot \int_{-\pi}^{\pi} \cos\theta \cdot P(\theta) \cdot \sin\theta \cdot d\theta$$
 (3)

9 Values for g range from -1 (backscattered radiation only) to 1 (forward scattered radiation10 only) in theory, and from 0 to 1 for particles in the atmosphere.

11 Corr et al. (2009) presented a review of studies estimating SSA at different wavelengths. For 12 the visible part of the spectrum, two different approaches have been presented. The first 13 (Dubovik et al., 2002), introduced sky radiance measurements in a matrix inversion technique 14 to calculate various aerosol microphysical properties. This methodology has been widely 15 applied in the AERONET. The second (Kassianov et al., 2005), proposed the use of radiative 16 transfer model (RTM) calculations, using as input measurements of AOD and the ratio of 17 direct to diffuse irradiance at specific wavelengths. However, in the case of SSA calculations 18 at UV wavelengths, enhanced measurement uncertainties, RTM input assumptions, and 19 interference of absorption by other gases (O<sub>3</sub>, NO<sub>2</sub>), make the retrieval more difficult. All 20 reported results concerning UV-SSA, utilize RTM combined with total and diffuse relative irradiance measurements (Herman et al., 1975; King and Herman 1979; King 1979; Petters et 21 22 al., 2003; Krotkov et al., 2005b; Corr et al., 2009; Bais et al., 2005) or absolute irradiance 23 measurements (Kazadzis et al., 2010; Ialongo et al., 2010; Bais et al., 2005). The review made 24 by Corr et al. (2009) also presents the major differences in the results of simulations of the 25 SSA, arising from RTM input assumptions, measurement techniques and retrieved 26 wavelengths. An additional problem is that previous studies have dealt with short time periods 27 due to the limited lifespan of experimental campaigns.

In this work, for the calculation of the UV-SSA, we adopt a methodology based on the idea of Krotkov et al. (2005a), Krotkov et al. (2005b) and Corr et al. (2009). The methodology, together with the retrieval tools used and technical assumptions made are presented in section

2. Results of UV-SSA measurements and their comparison with synchronous AERONET
 retrievals in the visible range are presented in section 3. Finally, discussion of the observed
 diurnal SSA patterns in Athens, SSA wavelength dependency as well as overall conclusions
 are presented in the last section of this work.

5

#### 6 2 Instrumentation and retrieval methodology

#### 7 2.1 Instrumentation

8 In this work we present estimates of SSA at two independently retrieved UV wavelengths 9 332nm and 368 nm for an urban site situated in Athens, Greece. The period of measurements 10 analysed is from July 2009 to May 2014. Since February 2009, the ground-based 11 Atmospheric Remote Sensing Station (ARSS) has been in continuous operation to monitor 12 ground radiation levels and aerosol loadings over Athens (Amiridis et al., 2009). ARSS is located on the roof of the Biomedical Research Foundation of the Academy of Athens (37.9 13 14 N, 23.0E, 130 m a.s.l.) (http://apcg.meteo.noa.gr/index.php?option=112&client=&langid=2) 15 and the campus is located near the city centre, 10 km from the sea (Gerasopoulos et al., 2009). 16 The horizon view is clear at 360 degrees viewing angle. ARSS is equipped with a CIMEL 17 CE318-NEDPS9 sun photometer for the retrieval of AOD at 8 wavelengths in the range of 18 340nm to 1640 nm, including polarization measurements as part of NASA's AERONET 19 (http://aeronet.gsfc.nasa.gov). The technical specifications of the instrument are given in 20 detail by Holben et al. (1998). ARSS is also equipped with an Ultraviolet Multi-filter 21 Radiometer (UVMFR) instrument for radiation measurements in the UV spectral region 22 (Harrison et al., 1994). UVMFR measures both total and diffuse irradiance at 7 specified 23 wavelengths (300, 305.5, 311.4, 317.6, 325.4, 332.4, and 368 nm) with a 2 nm nominal full 24 width at half maximum (FWHM) bandwidth. The instrument has been purchased on 25 November 2009 and the constructing company (Yankee Environmental Systems, USA) has 26 provided angular, spectral and absolute response functions of each wavelength channel of the 27 instrument that were measured at the National Institute of Standards and Technology (Figure 1). For the analysis included in this work we assume that the effective wavelengths for each 28 29 channel were stable during the whole period. Measurements of total and diffuse irradiance are 30 recorded every 10 seconds, and stored as 1 minute averages along with a computation of the 31 direct irradiance. Measurement data were angle-corrected, calibrated and analysed via the

- 1 YESDAS Manager software. The individually characterized cosine response, supplied with
- 2 each instrument, was used by system software to correct, in real time, for deviations from the
- 3 ideal cosine response (Harrison et al., 1994). For this work, we have used measurements of
- the two aforementioned instruments in conjunction with radiative transfer model (RTM) 4
- 5 calculations that have been performed using the Libradtran code (Mayer and Kylling, 2005).