# Peer review of "Aerosol absorption retrieval at ultraviolet wavelengths in a"

_Atmospheric Measurement Techniques, 2016_

## Referee Comment (RC1) · Anonymous Referee #1 · 8 Sep 2016

Review for Atmospheric Measurement Techniques

Title: Aerosol absorption retrieval at ultraviolet wavelengths in a complex environment

Authors: S. Kazadzis, P. I. Raptis, N. Kouremeti, V. Amiridis, A. Arola, E. Gerasopoulos, G.L. Schuster

General Comments: This paper presents some interesting results regarding the spectral variation of single scattering albedo from the visible into the UV wavelength region in Athens, Greece. Given the location's proximity to the Saharan desert there is an interesting variety of aerosol types including urban-industrial, desert dust and various mixtures of these. However there are several errors and/or confusing aspects of the paper that require modification or elaboration. One is the interpretation of Figure 5. When the 15 degree solar zenith angle data point is excluded then there is an obvious

trend of AERONET-UVMFR differences of AOD that range from ∼+0.02 to -0.01 as a function of solar zenith angle from 20 to 65 degrees, while the authors suggest there is no trend (page 12, lines 8-12). This suggests that the cosine response error in the UVMFR may not be fully accounted for, since the AERONET direct sun measurement of AOD with a narrow field of view does not have any solar zenith angle tendency. I also include several specific comments below that the author's need to address in order to correct or clarify some parts of the paper. One example is that Figure 13 as described in the text (page 22, lines 10-11) was missing from the review copy of this manuscript.

Specific Comments:

Page 2, lines 25-28: The authors suggest that AERONET almucantar retrievals are only for the visible (VIS) part of the spectrum. This is not true as the wavelengths that are input to the retrieval algorithm include two near-infrared wavelengths (870 and 1020 nm) in addition to two visible wavelengths (440 and 675 nm). Therefore the term "VIS-SSA" on line 28 is somewhat misleading.

Page 3, line 1: The word 'weaker' seems to be a poor choice of vocabulary here, perhaps 'difficult' would be better?

Page 3, line 5: Please include the fact that measurement accuracy is even more important here than measurement precision.

Page 3, line 5: Please note that the UV fraction of the energy in the total solar spectrum is very small and therefore it is not very important for radiative forcing estimates.

Page 3: Please include some mention in the introduction section of the satellite retrievals of SSA in the UV wavelengths as has been published in several papers by Omar Torres (GSFC). Include at least one Torres reference in this discussion.

Page 5, line 18: Please include "mid-visible" before SSA in this sentence.

Page 6, line 4: Note that aerosol SSA cannot be much lower than 0.2 due to particle diffraction effects.

[Figure]

Page 6, line 15-17: Note that Eck et al. (2003) also applied this approach to VIS-NIR wavelengths.

Page 7, line 25: Change 'constructing company' to 'manufacturer'

Page 8, line 15-16: Please give references here for the range of SSA, particularly for values as low as 0.5. This lower limit seems extreme to me.

Page 9, line 28: Please include the reference of Smirnov et al. (2000) for the AERONET cloud screening.

Page 12, Figure 4: Figure 4 is hard to read due to extreme compression of the y-axis and very small font for labels. The figure is also confusing since it implies an AOD of 1.5 from the UVMFR when the Cimel would measure only 1.0. The plots therefore appear to be inconsistent with the linear fit equations which have a slope of very close to 1.0. Please explain this apparent discrepancy.

Page 12, line 9: Figure 4 should be Figure 5 here.

Page 13, Figure 5: I assume that the 15-degree solar zenith angle has less observations than the other bins in Figure 5. It would be useful to show the number of data points that are included in each SZA interval bin.

Page 13, line 6: Please be clear that AERONET uses satellite derived climatological values for both ozone and NO2.

Page 13, line 17-18: However, this plot (Figure 6 right panel) suggests a cutoff of <65 degrees SZA for retrievals due since sensitivity decreases rapidly as SZA increases.

Page 15, line 10-11: Please note here that you have also applied the SZA restriction of >50 degrees, as shown in Figure 10 (on page 18). Also please note that the uncertainty in SSA from AERONET increases rapidly as AOD decreases, with uncertainty of ∼0.03 at AOD(440 nm)=0.5, see Dubovik et al. (2000; Table 4). Did you apply a minimum value of AOD to the AERONET and UVMFR retrievals of SSA shown in this paper?

The errors in SSA for the UVMFR shown in Figure 7 become very large at low AOD.

Page 16, Figure 8 caption: Please change 'the visible range' to 'at 440 nm'.

Page 16, line 23-24: I suggest also plotting the monthly mean AOD values to see if there is a relationship to the SSA retrievals.

Page 17, line 18-20: These should not be called error bars if they denote variability of SSA in each month. These 1-sigma values are likely due to both variation in aerosol properties and retrieval uncertainty. Please add a sentence to clarify this.

Page 18, Figure 10: Figure 10 shows the annual average diurnal variability. How does the diurnal variation change seasonally?

Page 18, Line 15: Please specify in the text the magnitude here for "higher AOD". Do you mean >0.4 at 440 nm?

Page 18, Line 15-17: This is a very confusing sentence about AERONET spectral dependence of SSA. Please rewrite or clarify this.

Page 18, Line 18: "at Washington" should be "near Washington, DC". Remember there is a Washington state on the west coast of the USA.

Page 20, Figure 12: Please use larger fonts for the labeling of Figure 12, it is currently very hard to read.

Page 20, Lines 9-11: What are the AOD levels for these low SSA cases with values <0.75? Are these very low SSA values from L1.5 or L2 retrievals? Please include this information in the text.

Page 20, Lines 15-21: "Russel et al (2010)" should be "Russell et al (2010)"

Page 21, Lines 2: I think that Figure 11 should be Figure 12 here.

Page 21, Lines 8: I think that Figure 12 should be Figure 13 here.

Page 21, Figure 13: There is no fine and coarse mode information given in the figure

13 although the text suggests it does have this information. Why are the January and December months missing in Figure 13. The caption of this figure says "in the lower plot" when there is only one plot panel. It seems as though the authors did not include all final figures in this manuscript!

Page 22, Lines 1-4: Why only scattering and not extinction for the AE?

Page 22, Lines 10-11: The Figure showing the temporal variability of AAE is missing from the review manuscript. This suggests poor quality checking of the manuscript by the authors.

Page 23, Figure 14: I suggest adding a fourth category to this plot : AE >1.2 to see the spectral SSA variation of fine mode cases.

Page 24, Line 20: What is a satellite post-correction validation result? What do you mean by post-correction?

Page 25, Line 1-3: It does not make any sense to me to compare this data to retrievals made in Washington DC without any further explanation. There is very little dust aerosol in Washington DC and therefore it would be expected for these two sites to differ significantly.

---

## Referee Comment (RC2) · N. A. Krotkov (Referee) · 19 Sep 2016

Review: S. Kazadzis et al., Aerosol absorption retrieval at ultraviolet wavelengths in a complex environment. (Plain text version. Full version - see PDF attachment.)

General comments The paper presents first long-term retrievals of the column average aerosol Single scattering Albedo (SSA) at two UVA wavelengths (332nm and 368nm) using UVMFR measurements of diffuse, global and direct surface irradiance in Athens, Greece.  The methodology retrieving SSA using UVMFR direct to global irradiance Ratio (DGR) measurements has been previously developed and authors appropriately reference previous works. The paper would benefit from adding more details of UVMFR operations (e.g., picture of the site, procedures for cleaning Teflon diffuser, checking diffuser alignment, night-time bias correction, aureole correction).

[Figure]

The paper presents new results that will be of interest to broad aerosol, atmospheric composition and air quality communities. This paper first analyzes long-term (5 year) UVMFR SSA retrievals that allows for statistically robust analysis of the absorbing aerosol climatology in Athens. Long-term comparison with standard AERONET SSA inversions at 440nm is also new. The paper further uses standard AERONET retrievals of the column effective imaginary refractive index in VIS – NIR wavelengths and methodology of Schuster et al., (2016) to calculate volume fractions of absorbing aerosol components (e.g., black and brown carbon, hematite, goethite) in Athens. This is new and interesting development. The main result (Fig.13) is that Brown Carbon dominates column aerosol absorption. October peak of BrC volume fraction is interesting new result. Authors should try to explain their result, e.g., using in-situ measurements. The possible enhancement of this new approach would be extending Schuster et al., (2016) methodology by using UVMFR retrieved imaginary refractive indices at 332nm and 368nm [Krotkov et al., 2005b].

The paper is appropriate for publishing in AMT after important technical corrections, such as adding missing references (e.g., IPCC 2007, 2013, 2014) and missing figure (p22,L10-11: "Figure 13 shows the temporal variability of AAE(440-870) and AAE(332-440)." ). The manuscript also needs language and punctuation improvements and will benefit from editing by native English speaker.

I recommend publishing the paper after corrections and implementing additional technical suggestions described below.

Use consistently italic or regular font in references and examples. Use dot and comma after abbreviations, such as "e.g., et al." Do not use apostrophes in SSAs , AEs and AODs All references need consistent formatting according to AMT style.

Specific suggestions: 1,18: properties -> property ( Only one absorption property, i.e., SSA is retrieved). 1,19 5-years period 1,22: and study absorption spectral behavior of the [retrieved] SSA values 1,24: towards lower shorter UV wavelengths 1,25: High
Strong SSA wavelength dependence ... 1,27: "SSA decrease with decreasing extinction optical depth, suggesting an effect of the different aerosol composition" – this could be due to in increased SSA uncertainties at lower AODs

2,2: "were investigated to understand seasonal variability of the results" – to explain? 2.6. e.g., 2.7: IPCC references are missing 2,8: "... as it appears that climate change is accelerating with aerosols impacting ..." - This sentence needs clarification and reference: how aerosol and climate changes are related? 2,13: "significant aerosol absorption uncertainties in [modeled?] global Single Scattering Albedo (SSA)," 2,16: mixture [mixing state?] 2,20: 50% change [decrease?] in the [surface] erythermal irradiance 2,21: et al., 2,22: e.g., (add comma) 2,24 "a fixed irradiance path" – please, re-word. Surface irradiance is a result of averaging different direct and scattered photons arriving at the surface via different paths through the atmosphere. 2,27: Do not use italic font in references. 2,28: VIS-SSA -> column average SSA retrievals at the visible and near IR wavelengths (i.e., 440nm, 670nm, 870nm, 1020nm).

2.29-2.32: "In addition, Goering et al. (2005), Taylor et al (2008) and Kudo et al. (2008) have proposed estimation techniques for the retrieval of spectral aerosol optical properties by combining multi-wavelength measurements using a priori constraints that are applied differently than in the single wavelength methods."

Suggest replacing this sentence with:

In addition, surface direct and diffuse irradiances had been used to derive spectral AOD and SSA at visible and UV wavelengths (King and Herman 1979; King 1979; Petters et al., 2003; Eck et al., 1998; Krotkov et al., 2005b; Bais et al., 2005; Goering et al., 2005; Taylor et al., 2008; Kudo et al., 2008; Corr et al., 2009).

2,32: "SSA retrieval in the ultraviolet (UV) part of the spectrum is weaker with large uncertainties." – I suggest removing this sentence.

3,10: " ...like organic, nitrate and aromatic aerosols are still poorly known" add references, e.g., Jacobson, M. Z. (1999), Isolating nitrated and aromatic aerosols and nitrated aromatic gases as sources of ultraviolet light absorption, J. Geophys. Res., 104, 3527–3542.

3,16-18: Barnard et al. (2008) [ and Corr et al., (2009) ] ...in a case [field] study ... found that, in the near-UV spectral range (250 300 to 400 nm) 3,28: "in at 870 [nm] " "could be a hint reason for ..." 3,32: " using Brewer [direct sun and global irradiance spectral UV ] measurements ..."

4,2: and They used imaginary refractive index ... and found ... 4,10: (e.g. Zerefos et al., 2012; - add more references 4,11: "comparable in magnitude [or exceeding ] with those caused by the decline in stratospheric ozone [depending on wavelength] 4,13 "reduction of 7% of AOD ..." - at what wavelength? 4,17: "tropospheric photochemistry[, causing:]

5,5: "Solar irradiance satellite retrieval algorithms are directly affected " -> satellite retrieval algorithms of the surface UV irradiance are directly affected ... 5,8: absent from [current] satellite (e.g., OMI) retrieval algorithms 5,12: "Uncertainty on [in] commonly used .." 5,13: "... fall short in precision due to large uncertainties in the input parameters ..." -> The model accuracy of the surface UV irradiance is limited by large uncertainties in the input parameters ...

5,24- 6,10 – suggest deleting this paragraph as common knowledge. Move Equation (2) to the beginning of section 2.2

6,11-27 – this paragraph looks repetitive and could be blended with the earlier part of introduction. 7,25: constructing [manufacturing] company 7,29: irradiance[s]

8,5: "...in conjunction with radiative transfer model (RTM) calculations that have been performed using the Libradtran code (Mayer and Kylling, 2005). " 8,12: "SSA is a key aerosol optical property and describes the portion of solar irradiance that is scattered from the main direct beam passing through the atmosphere." – Equation (2) can be

moved after this sentence.

9,4: "raw voltage measurements [corrected for night-time voltages and non-ideal angular response] could be used." 9,19: "dt." -> time interval

10,fig 2 caption "...for a day with variable cloudiness [in the afternoon] 10, 9 : "... for performing [determining] extraterrestrial Langley calibration constant (ETC) determination ... 10,10: "the Beer-Lambert law for to the UVMFR direct [voltage] measurements" 10,13-14: "extrapolated AOD at UVMFR wavelengths" – Clarify how AOD was extrapolated? 11,6: AOD's at 332 nm and 368 were ... 11, 15: extrapolation [using ln(AOD) versus ln(wavelength) ]? 12, Fig 4 caption: "Comparison of CIMEL and[extrapolated] and ... UVMFR retrieved AODs for ... 332 nm (up top panel) and 368 nm (down bottom panel)." 12,9: "as a function of SZA (figure 5 4)." 12,12: "...are included [found] ..." 12,18: " ...due to instrumental (filter related) changes ..." – Most likely reason for ETC change is UVMFR Teflon diffusor contamination. Explain how often the UVMFR diffusor was cleaned and what cleaning procedures applied?

12, 19 AODs 's deviations ... errors on in SSA calculations .. 13,6: were deployed for the use of [used for construction of] the LUT ... 14, Fig 6. Caption Figure 6 LUT of direct to global ratio at 368nm ... color bar represents assumed SSA values.

14,13: average annual monthly AODs 's 15,12: SSAs 's 15,15: role on [in] the uncertainty 15,21: a [close] match between 15,22: "This range broadens at low SZA and low aerosol level cases" – Please, clarify this sentence and refer to Figs 5 and 6.

15, 26: "AOD uncertainty is considered as $\pm2\%$ for 368nm and $\pm4\%$ for 332nm," Should it be absolute AOD uncertainties: 0.02 at 368nm and 0.04 at 332nm ? 16,3 AODs 16,5: In the same figure, 16,6: mean AODs 's [for each SZA bin] and $1\sigma$, the error bars equal to one standard deviation are shown 16,16: Figure 8 Daily Mean daily SSAs ... 16,20: for 332nm (368nm)" - Should it be reversed, i.e. , at 368nm (332nm) (SSA at 332 nm is generally lower than at 368nm) – fig 7. ?

17, Fig. 9 : Include X axis title. Are spectral differences between SSA at 368nm and at 3232nm between November and March statistically significant? Apply standard statistical significance tests. 17,1: for the specific area," – correct reference Pareskevopoulou et al (2014) -> Paraskevopoulou, et al., 17,2: at in February and November 17, 3 at in a 5 year (2008-2013) . . . 17,4: have similar behavior SSAs . . . 17,16-17: "for different SSA uncertainty bins according to analysis of in the previous section . . .

18, Figure 10: Suggest local time or SZA as X axis 18, Figure 10 caption: Mean values per hour plotted [with error bars showing one standard deviation] at $1\sigma$ 18,8: are [also] linked . . . 18,8: derived at AERONET calibration site in Greenbelt, Maryland USA Washington

19,15: link between [SSA] wavelength dependence and

20, 5 The results of in figure 12 20, 9 . . .relatively higher than SSA440) tend [to occur] towards high AEs . . . 20,10 attributed in [to] polluted . . . 21,6: Schuster et al., (2016) 21,7: method separates [contributions from] black carbon, organic carbon, hematite and goethite, using [to the retrieved] refractive index . . . 21,9: 8-9: "Figure 12 shows the fractions of total aerosol [column] volume attributed to these components in both fine and coarse mode . . ." – This should be Fig. 13. There are no fine and coarse mode fraction data in Fig.13.

21,14: ". . . higher at [in October] OCTOBER " – It will be interesting to explain BrC peak in October compared to other months. Are there in-situ measurements in Athens that could support this finding?

21,17: Figure 13 caption: "Volume fraction[s] (in the lower plot) of absorbing aerosol components . . ."

22,4: " for atmospheric aerosol [mixtures] scattering varies . . ." 22,10-11 "Figure 13 shows the temporal variability of AAE(440-870) and AAE(332-440)." – This figure (14?) is missing.

23, Figure 14 – This should be figure 15 23, 7. I suggest adding new reference, which shows previously measured AERONET-UVMFR SSA spectral dependence in Thessaloniki, Greece:

N. Krotkov ; G. Labow ; J. Herman ; J. Slusser ; R. Tree ; G. Janson ; B. Durham ; T. Eck ; B. Holben; Aerosol column absorption measurements using co-located UV-MFRSR and AERONET CIMEL instruments. Proc. SPIE 7462, Ultraviolet and Visible Ground- and Space-based Measurements, Trace Gases, Aerosols and Effects VI, 746205 (August 20, 2009); doi:10.1117/12.826880.

23,19: for all SZA[s]

24,10: "We have also [used] the produced dataset to investigate . . . 24,18: "We expect a possible decrease in specific days/cases of regional O3 due to the enhanced aerosol absorption" - This conclusion is not supported in the main text. Add Chemical model results to support this.

Please also note the supplement to this comment:
http://www.atmos-meas-tech-discuss.net/amt-2016-273/amt-2016-273-RC2-supplement.pdf

---

## Author Comment (AC1) · 15 Nov 2016

We would like to thank the reviewer for the comments and recommendations.

We apologize for the fact that the uploaded version was not the absolute final. Due to that, there were several points mostly related with the AAE figure not showed and figure 13 that are not correct and have been revised through the reviewer's suggestions.

In this document we have included the original questions and recommendations in italics and our responses.

General comments

*One is the interpretation of Figure 5. When the 15 degree solar zenith angle data point is excluded then there is an obvious trend of AERONET-UVMFR differences of AOD that range from +0.02 to -0.01 as a function of solar zenith angle from 20 to 65 degrees, while the authors suggest there is no trend (page 12, lines 8-12). This suggests that the cosine response error in the UVMFR may not be fully accounted for, since the AERONET direct sun measurement of AOD with a narrow field of view does not have any solar zenith angle tendency.*

We have included the following sentence in the manuscript:

"An AOD, SZA dependent trend, in the order of 0.02 (if excluding the $15^o$ SZA bin) can be observed which could be attributed to ETC determination uncertainty or non ideal correction for the cosine response error of the UVMFR."

However, since we are mainly interested on the possible effects of the different AODs in the SSA retrievals from both UVMFR and CIMEL, we tried to investigate this effect.

So we used the radiative transfer code for the SSA-UVMFR retrieval with inputs:

a. UVMFR derived AODs at 368nm and 332nm

b. The synchronous AERONET (interpolated to 368nm and extrapolated at 332nm) AODs.

[Figure]

Figure: SSA retrieval at 332nm and 368nm using the UVMFR AODs (XX' axis) and the CIMEL AODs (YY' axis)

The results show the low impact of the AOD differences in the retrieved UVMFR SSAs, even including calibration and UVMFR angular correction related uncertainties plus the uncertainties connected with the interpolation/extrapolation of the CIMEL AOD wavelengths to the UVMFR ones.

***Page 2, lines 25-28:  The authors suggest that AERONET almucantar retrievals are only for the visible (VIS) part of the spectrum.   This is not true as the wavelengths that are input to the retrieval algorithm include two near-infrared wavelengths (870 and 1020 nm) in addition to two visible wavelengths (440 and 675 nm). Therefore the term "VIS-SSA" on line 28 is somewhat misleading.***

The error has been corrected in this paragraph, and at all other parts of the text, to include both visible and near infrared regions of spectrum when referring to the almucantar retrievals.

***Page 3, line 1: The word 'weaker' seems to be a poor choice of vocabulary here, perhaps 'difficult' would be better?***

The sentence has been restated.

***Page 3, line 5:  Please include the fact that measurement accuracy is even more important here than measurement precision.***

Accuracy is way more important than precision for our retrieval and the sentence has been restated.

*Page 3, line 5: Please note that the UV fraction of the energy in the total solar spectrum is very small and therefore it is not very important for radiative forcing estimates.*

Comment was mentioned in the text

*Page 3: Please include some mention in the introduction section of the satellite retrievals of SSA in the UV wavelengths as has been published in several papers by Omar Torres (GSFC). Include at least one Torres reference in this discussion.*

Two sentences has been added on the aspect of SSA satellite retrieval in the UV and a O. Torres overview of OMI retrievals has been referred:
"Torres et al., (2007), in an overview study of OMI aerosols products, summarized the algorithmical techniques of SSA satellite retrieval at 388nm, which uses spectral variability between 354nm and 388nm , 388nm reflectance and a selection on the aerosol type. They compared to AERONET SSA at 440nm and found a root mean square error of 0.03."

*Page 5, line 18: Please include "mid-visible" before SSA in this sentence.*

It has been changed in reviewed manuscript.

*Page 6, line 4: Note that aerosol SSA cannot be much lower than 0.2 due to particle diffraction effects*

The sentences were changed to:
"Theoretically SSA values range from 0 (totally absorbing aerosols) to 1 (totally scattering aerosol). Actual SSA values in the atmosphere can be found in the range of 0.6 to 1."

*Page 6, line 15-17: Note that Eck et al. (2003) also applied this approach to VIS-NIR wavelengths.*

A reference to this work has been added.

*Page 7, line 25: Change 'constructing company' to 'manufacturer'*

The sentence has been restated.

*Page 8, line 15-16: Please give references here for the range of SSA, particularly for values as low as 0.5. This lower limit seems extreme to me.*

We provided the Corr et al., 2009 reference reporting values from 0.6 to 1 and changed the limits. However AERONET L1.5 data can be found down to 0.5. But since they are related with low AODs can be considered highly uncertain.

*Page 9, line 28: Please include the reference of Smirnov et al. (2000) for the AERONET cloud screening.*

Reference included.

*Page 12, Figure 4: Figure 4 is hard to read due to extreme compression of the y-axis and very small font for labels. The figure is also confusing since it implies an AOD of 1.5 from the UVMFR when the Cimel would measure only 1.0. The plots therefore appear to be inconsistent with the linear fit equations which have a slope of very close to 1.0. Please explain this apparent discrepancy.*

The figure axis were corrected. Fit equations were actually correct. Figure was re-shaped in order to be easier to read. We have revised the figure as follows:

[Figure]

Figure: Comparison of CIMEL and UVMFR retrieved AODs for synchronous measurements for 332 nm (left panel) and 368 nm (right panel).

*Page 12, line 9: Figure 4 should be Figure 5 here*.

Corrected

*Page 13, Figure 5: I assume that the 15-degree solar zenith angle has less observations than the other bins in Figure 5. It would be useful to show the number of data points that are included in each SZA interval bin.*

The bar plot above shows data sampled in each bin. The 15º bin has less points but ~7000 data points is still a statistically large collection of data.

[Figure]

Figure: Number of data per SZA bin

*Page 13, line 6: Please be clear that AERONET uses satellite derived climatological values for both ozone and NO₂.*

The sentence has been restated.

*Page 13, line 17-18: However, this plot (Figure 6 right panel) suggests a cutoff of <65 degrees SZA for retrievals due since sensitivity decreases rapidly as SZA increases.*

Yes we agree that is why we have used measurements for solar zenith angles lower than 70 degrees in this work.

*Page 15, line 10-11: Please note here that you have also applied the SZA restriction of >50 degrees, as shown in Figure 10 (on page 18). Also please note that the uncertainty in SSA from AERONET increases rapidly as AOD decreases, with uncertainty of 0.03 at AOD (440 nm)=0.5, see Dubovik et al. (2000; Table 4). Did you apply a minimum value of AOD to the AERONET and UVMFR retrievals of SSA shown in this paper? The errors in SSA for the UVMFR shown in Figure 7 become very large at low AOD*
.
The enhanced cimel derived, dataset we used is as described the L1.5, with the added criterion, that L2 size distribution is available. So there are data below 50º. In our calculations we did not filter data due to low AOD. Obviously measurements at very low AODs, have high uncertainty at SSA retrieval, as shown in figure 7, and it is visible at figure 11 were SSA values for low AODs are largely scattered. However, for all comparisons with AERONET measurements (figures 12, 14) these measurements are filtered, as it cannot pass the criterion for enhanced L1.5 inversion retrievals, and only synchronous to those UVMFR are used for these statistics.
We have added the sentence: 'We have to note that since no restriction has been introduce for UVMFR SSA retrievals at low AOD's the SSA uncertainty related with these data becomes larger as seen also in figure 7.'

*Page 16, Figure 8 caption: Please change 'the visible range' to 'at 440 nm'.*

It has been changed.

*Page 16, line 23-24:  I suggest also plotting the monthly mean AOD values to see if there is a relationship to the SSA retrievals.*

[Figure]

We have added the monthly mean AOD on top of the SSA monthly means.

Sentence added: "Looking at the monthly mean AODs; despite the fact that standard deviations of both SSA and AOD's are large, it can be seen that higher AOD's are associated with less absorbing aerosol cases."

*Page 17, line 18-20: These should not be called error bars if they denote variability of SSA in each month.  These 1-sigma values are likely due to both variation in aerosol properties and retrieval uncertainty. Please add a sentence to clarify this.*

The text has been changed to:
"However, the statistical one standard deviation bars are quite large. These bars describe the variability of the SSA's during each hourly bin, but also include the retrieval uncertainty."

*Page 18, Figure 10: Figure 10 shows the annual average diurnal variability. How does the diurnal variation change seasonally?*

Analyzing different seasons, there is no evidence that the pattern changes for different seasons. Differences are well within the statistical standard deviation bars. Having a look at the SZA dependence of the SSA at 368nm and 332nm it does not seem to have any clear dependence too.

[Figure]

Figure: SSA at 332 nm and 368 nm and 1 standard deviations for 5 degree SZA bins

*Page 18, Line 15: Please specify in the text the magnitude here for "higher AOD". Do you mean >0.4 at 440 nm?*

A threshold value of 0.6 has been added to the manuscript to clarify the sentence

*Page 18, Line 15-17: This is a very confusing sentence about AERONET spectral dependence of SSA. Please rewrite or clarify this.*

The sentence has been revised.

*Page 18, Line 18: "at Washington" should be "near Washington, DC". Remember there is a Washington state on the west coast of the USA.*

Corrected: "CIMEL retrievals show an almost constant value of the SSA ~0.92, while lower values have been retrieved at smaller AODs . Similar results were reported by Krotkov et al. (2005b) when analyzing measurements derived at at AERONET calibration site in Greenbelt, Maryland USA."

*Page 20, Figure 12: Please use larger fonts for the labeling of Figure 12, it is currently very hard to read.*

Font size has been changed.

*Page 20, Lines 9-11:  What are the AOD levels for these low SSA cases with values <0.75? Are these very low SSA values from L1.5 or L2 retrievals? Please include this information in the text.*

The 0.7 value refers to the Angstrom Exponent and not the SSA. All values in this paragraph refers to differences among 440nm and 368nm.

*Page 20, Lines 15-21: "Russel et al (2010)" should be "Russell et al (2010)"*

Done

*Page 21, Lines 2: I think that Figure 11 should be Figure 12 here.*

Corrected

*Page 21, Lines 8: I think that Figure 12 should be Figure 13 here.*

Corrected

*Page 21, Figure 13: There is no fine and coarse mode information given in the figure 13 although the text suggests it does have this information. Why are the January and December months missing in Figure 13.*

We have revised the figure, including the original one in this manuscript version.

[Figure]

[Figure]

The reason for January/December having no data is related to two thresholds applied here. Only data when AOD@440nm >= 0.2 was included and additionally only months when there was at least 10 retrievals. Both had influence that these two months got excluded, e.g. in December there were only three cases of retrievals with AOD larger than 0.2.

*Page 22, Lines 1-4: Why only scattering and not extinction for the AE?*

Corrected to aerosol extinction

***Page 22, Lines 10-11: The Figure showing the temporal variability of AAE is missing from the review manuscript. This suggests poor quality checking of the manuscript by the authors.***

We apologize for that. For some reason not the absolute final version of the paper has been uploaded. Due to that, there are several points related with the AAE figure not showing here and the figure 13 that are not correct and have been revised through the reviewer's suggestions.

***Page 23, Figure 14: I suggest adding a fourth category to this plot : AE >1.2 to see the spectral SSA variation of fine mode cases.***

We have included similar data for AE>1.2 as suggested.

[Figure]

Figure: Wavelength dependence of SSA from synchronous CIMEL and UVMFR measurements. Blue points represent all data points, red data retrievals with AOD>0.2, green points data with AE>1.2 and black data only dust aerosol cases. Vertical bars represent 1 standard deviation of the calculated mean.
Comment added: "Fine mode cases show smaller spectral dependence ($SSA_{440nm}$-$SSA_{332nm}$ <0.03)."

***Page 24, Line 20: What is a satellite post-correction validation result? What do you mean by post-correction?***

We added the Arola et al., 2009 reference. We mean that satellite products like OMI satellite related solar UV has been proven that have to be corrected for absorbing aerosols. So as post-correction we mean a factor that has been applied on the satellite results. We have erased the word validation from this sentence as it is wrong.

***Page 25, Line 1-3: It does not make any sense to me to compare this data to retrievals made in Washington DC without any further explanation. There is very little dust aerosol in Washington DC and therefore it would be expected for these two sites to differ significantly.***

**We write:**
The extended SSA dataset …provides additional information on the effect of varying background aerosol conditions and higher aerosol absorption, than that provided by Washington, DC, where dust aerosol cases are very rare.

So we are just mentioning that the dataset used here provides information on partly different aerosol conditions than the ones in Washington DC. There are a lot of days with similar aerosol conditions for the two cities, but also others associated with dust aerosol intrusions that enhance our understanding for SSA@UV wavelengths on such conditions.

**Aerosol absorption retrieval at ultraviolet wavelengths in a complex environment**

**S. Kazadzis[1,2], P.I. Raptis[2], N. Kouremeti[1], V. Amiridis[3], A. Arola[4], E. Gerasopoulos[2], G.L. Schuster[5]**

[1]{Physikalisch-Meteorologisches Observatorium Davos, World Radiation Center (PMOD/WRC) Dorfstrasse 33, CH-7260 Davos Dorf, Switzerland}

[2] {Institute of Environmental Research and Sustainable Development, National Observatory of Athens, Greece}

[3]{Institute of Astronomy Astrophysics, Space Applications and Remote Sensing, National Observatory of Athens, Greece}

[4] {Finnish Meteorological Institute, Kuopio Unit, Finland}

[5] {NASA Langley Research Center, Hampton, VA, USA}

Correspondence to: stelios.kazadzis@pmodwrc.ch

**Abstract**

We have used total and diffuse UV irradiance measurements with a multi-filter rotating shadow-band radiometer (UVMFR), in order to calculate aerosol absorption  property (Single Scattering Albedo - SSA) in the UV range, for a  5-years period in Athens, Greece. This data set was used as input to a radiative transfer model and the SSA for 368nm and 332nm has been calculated. Retrievals from a collocated CIMEL sun-photometer were used to validate the products and study absorption spectral behavior of retrieved SSA values at these wavelengths. UVMFR SSA together with synchronous,CIMEL-derived, retrievals at 440nm, show a mean of 0.90, 0.87 and 0.83, with lowest values (higher absorption) towards  shorter wavelengths. In addition, noticeable diurnal variations of the SSA in all wavelengths are revealed, with amplitudes in up to 0.05.  Strong SSA wavelength dependence is found for cases of low Ångström exponents and also an SSA decrease with decreasing extinction optical depth, suggesting an effect of the different aerosol composition.

However, part of this dependence, for low aerosol optical depths, is masked by the increased SSA retrieval uncertainty. Dust and Brown Carbon UV absorbing properties were investigated to  explain seasonal variability of the results.

**1    Introduction**

[revised manuscript text omitted]

**2.2  Retrieval methodology**

SSA is a key aerosol optical property and describes the portion of solar irradiance that is scattered from the main direct beam passing through the atmosphere. Changes in SSA influence mostly the diffuse radiation reaching the earth's surface, while its effect on direct radiation can be considered negligible.  SSA at a wavelength λ provides the contribution of aerosol particle scattering relative to the total extinction (absorption plus scattering),

$$SSA = \frac{b_{sca}(\lambda)}{b_{abs}(\lambda) + b_{sca}(\lambda)} \tag{1}$$

Theoretically  SSA values range from 0 (totally absorbing aerosols ) to 1 ( totally scattering aerosol). Actual SSA values in the atmosphere can be

found usually in the range of  0.5 to 1 (Corr et al., 2009). The asymmetry parameter, is the phase function (P) weighted average of the cosine of the scattering angle (θ) over all directions. Assuming azimuthal symmetry, the scattering angle integration extends from – π to +π such that the asymmetry parameter (g) is given by

$$g = \frac{1}{2} \cdot \int_{-\pi}^{\pi} cos\theta \cdot P(\theta) \cdot sin\theta \cdot d\theta \tag{2}$$

Values for g range from -1 (backscattered radiation only) to 1 (forward scattered radiation only) in theory, and from 0 to 1 for particles in the atmosphere.

Model calculations can be used for retrieving SSA when global and/or diffuse spectral irradiance, solar zenith angle (SZA), total column ozone, and AOD are known (Krotkov et al., 2005b; Kazadzis et al., 2010; Ialongo et al., 2010; Corr et al., 2009; Bais et al., 2005). In our retrieval methodology we have used partly the basic approach that is described in detail in the Corr et al. (2009), Krotkov et al. (2005a) and Krotkov et al. (2005b). This approach consists of measurements of the direct to global irradiance ratios (DGR) and AODs measured with the UVMFR instrument for our case, that are used as basic input parameters to the RTM for the calculation of the SSA at 332nm, and 368nm. These wavelengths are selected for having the lowest ozone absorption from the seven available (Bass and Paur ,1985). The advantage of this method is that the same detector and filter measure global and direct irradiance, thus there is no need for absolute irradiance calibration and raw voltage measurements –-corrected for nighttime voltages and angular response - could be used.

Global irradiance measurements from the UVMFR have been used in order to distinguish cloud free conditions for each of the one minute measurements. Clouds are detectable in the measured UVMFR global irradiance (GI) (at 368nm) since they cause larger variability than aerosols. For distinguishing between cloudy and cloud free conditions, we have applied an updated version of the method of Gröbner et al., (2001). The method is based on the comparison of the measured global irradiance with radiative transfer calculations for cloud free conditions and quality assurance is checked by the following criteria:

a. The measured GI has to lie within the modeled (cloud free) GI for a range of aerosol loads (AOD at 500 nm of 0.1 and 0.8, respectively), corresponding to the 5th and 95th percentile of the AERONET data for the examined location and period

b. The rate of change in the measured GI with SZA has to be within the limits depicted by the modeled cloud free GI, otherwise the measurements are assumed cloud contamination.

c. All measured GI values within a time window (dt= ±10 min) should be within 5% of the modeled cloud free GI, and adjusted to the level of the measurement, using an integral over time interval.

If at least 85% of the points in dt pass tests a) – c), then the central point is flagged as cloud free. In this study, we have allowed a tolerance level of ±10% for tests a) and b) in order to compensate for differences between the modeled GI and measured GI due to instrumental uncertainties, as well as for usage of average climatological parameters (constant total ozone column, SSA, e.t.c.) as inputs to the model. We have limited the method to SZA<70º to avoid uncertainties related with low solar irradiance levels. An example of the results of the method is presented in figure 2 for a day with variable cloudiness. It has to be noted that in all CIMEL-UVMFR comparisons, using synchronous measurements, both the above method and AERONET cloud  screening algorithm (presented by Smirnov, et al, 2000) are taken into account.

[Figure]

**Figure 2**. Determination of cloudless 1-minute measurements (red), from all measurements (blue) for a day with variable cloudiness in the afternoon.

Measurements of the diffuse and global irradiance from the UVMFR have been used in order
to retrieve the direct irradiance at 332nm and 368nm.  We used the AERONET database to
select days with very low AOD (<0.1). For the urban environment of Athens such cases are
related with the presence of northern winds. Afterwards we selected cloudless sky half-days
for  determining extraterrestrial Langley calibration constant (ETC)
by applying the Beer-Lambert law  UVMFR direct voltage measurements. $V_{olangley}$ in
figure 3 represent the half day values calculated with this method. In order to examine the
consistency of this approach we calculated the $V_{0cimel}$ also as

$$V_{0\,cimel} = V e^{\mu\,(AOD_{cimel}+\tau_{rayleigh})}$$

where V is the voltage measured by UVMFR,   $\mu$ is the air mass, $AOD_{cimel}$ is the extrapolated
AOD at UVMFR wavelengths and $\tau_{rayleigh}$ is Rayleigh scattering optical depth. Daily averages
of $V_{0cimel}$ for the selected days were compared with $V_{0langley}$ as presented at figure 3. These
independent approaches appear stable through the years, with no obvious drift or change, so
we decided to use a single ETC for the whole period for each wavelength.

[Figure]

**Figure 3**. ETC values at 368nm, calculated using Langley plots of UVMFR measurements,
and Using Cimel extrapolated AOD's as input, for selected (low AOD's and clear sky) days
for the whole period

AOD's at 332 nm and 368 were calculated using the selected UVMFR derived ETC. In
contrast with the Krotkov et al., 2005a approach we have not transfered the CIMEL ETCs to

the UVMFR measurements; rather, we have independently calculated UVMFR-based AODs. Validation of the results was performed based on synchronous UVMFR and CIMEL measurements. The mean AOD calculated from the 1 minute UVMFR measurements within ±5 minutes from the CIMEL measurement (when the UVMFR 10 minute period is characterized by cloudless conditions) has been defined as synchronous. Since the CIMEL instrument provides measurements of AOD at 340 nm and 380 nm, we first calculated the CIMEL derived AOD at 332 nm and 368 nm,  applying least square quadratic spectral extrapolation, using ln(AOD) as function of ln(wavelength) from AERONET measurements at 340nm 380nm, 440nm and 500nm.  (Eck et al, 1999).

[Figure]

**Figure 4**. Comparison of CIMEL and UVMFR retrieved AODs for synchronous measurements for 332 nm (left panel) and 368 nm (right panel).

The results of this comparison have a Pearson product moment correlation coefficient equal to 0.96 and 0.98 respectively for 332nm and 368nm AODs. Mean differences were zero, with standard deviations of 0.031 and 0.025 for the respective wavelengths, comparable with the CIMEL AOD retrieval uncertainty of ±0.02. The quality of the data produced can be verified by comparing the AOD's retrieved by the two instruments as a function of SZA (figure 5). Relative stability of the AOD differences (that are in the order of the AERONET uncertainties), verifies the validity of the calibration of the UVMFR AOD's.  found in this procedure. An AOD, SZA dependent trend, in the order of 0.02 (if excluding the $15^{o}$ SZA bin) can be observed which could be attributed to ETC determination uncertainty or non ideal correction for the cosine response error of the UVMFR.

In figure 5, AOD's have been grouped in bins of 5 degrees (of SZA). The differences shown in figure 5 include ETC determination accuracy, the extrapolation of CIMEL AOD at 368nm, together with instrumental/measurement errors. Using a single UVMFR ETC for the whole period provides very good agreement between the two instruments. However, this may not be the case for all UVMFR instruments using this approach as ETC may suddenly or gradually change especially for years-long time series due to instrumental (filter related) changes. AOD's deviations could lead to large errors  in SSA calculations, so this comparison ensures that these errors are minimized. The instrument's teflon diffuser contamination is the most common reason for long term changes in the ETC. Maintenance procedure for the Athens instrument included cleaning and inspection of the diffuser and check of the levelling and shadowing, three times a week. In addition, metal spikes have been built around the instrument to avoid the diffuser destruction by birds.

[Figure]

2    **Figure 5** AOD differences between CIMEL and UVMFR at 368 nm, as a function of solar

3                                  zenith angle.

4    We calculated look up tables (LUT) with the RTM, of DGR at 368nm and 332nm as a

5    function of SZA, AOD, SSA, asymmetry factor (g) and total column ozone.

6    CIMEL/AERONET mean daily ozone values and climatological –satellite derived NO$_2$

7    values were used for construction of the LUT while for g, we used the

8    mean daily value as retrieved at 440nm from the CIMEL instrument measurements when

9    available and the mean value of the whole period equal to 0.7 (2σ standard deviation of the g

10   during this period was 0.04) otherwise. Using UVMFR AOD and DGR measurements, we

11   then calculated the matching SSA values for each individual UVMFR DGR measurement.

12   LUT examples are visualized in figure 6, for clarification of the method. For known SZA and

13   AOD (in cloudless sky conditions), the variability of the DGR is caused by aerosol properties

14   other than AOD. At low aerosol loads this variation is nearly negligible, but it becomes more

15   important at higher aerosol load. More absorbing aerosols lead to smaller values of DGR.  It

16   is crucial to observe the range of SSAs in the two examples. For low AOD's, accurate SSA

17   determination requires very low uncertainty of the DGR and the AOD measurement. While

18   for high AOD's the range of DGRs for a particular SZA is quite large.

[Figure]

**Figure 6** LUT of direct to global ratio at 368nm, as calculated for AOD 0.1 (left) and 0.8 (right) with respect to SZA (g=0.7), colourbar represents assumed SSA values.

**2.3 Retrieval Uncertainties**

The CIMEL sunphotometer provides SSA inversion retrievals characterized as Level 1.5 and Level 2.0 data. Level 2.0 (L2) data are recommended by AERONET as they have less uncertainty but are restricted in measurement to SZA>50 degrees, AOD at 440 nm> 0.4 and homogeneous sky conditions. These limitations make AERONET SSA L2 worldwide measurements unsuitable for:

a. climatological studies due to the AOD restriction that limits analyses only to areas having large average annual AOD's, or to cases of moderate to high aerosol episodes in specific areas. As an example, for the urban site of Athens, which is one of the most polluted cities in Europe, the number of measurements is limited to an average 11 cases per month for the whole analysis period.

b. diurnal variation studies due to the SZA restriction. For mid and low latitude sites, this limitation leads to a severe lack of information on diurnal SSA patterns as there are only few wintertime measurements and close to zero measurements at local noon.

[Figure]

**Figure 7** Uncertainty estimate (color) for the SSA retrieval from UVMFR as a function of AOD and solar zenith angle for 368nm (left panel) and 332nm (right panel), based on DGR and AOD uncertainties. Superimposed, mean AODs for 2.5 degree bins of solar zenith angle are shown.

Level 1.5 data: AERONET Level 1.5 (L1.5) SSA data are provided by AERONET for all AOD's and at all SZA that almucantar scans are performed. In this work L1.5 data were used, but with an extra quality control. We have ignored SSA L1.5 data when L2 size distribution is not available. Thus we have an enhanced L1.5 SSA data set with AOD<0.4, but with L2 cloud screening, calibrations and quality controls. Data has been compared with UVMFR retrieved SSA's taking into account limitations related with the retrieval uncertainties. Khatri et al (2016) studied AERONET SSA retrieval uncertainties, in order to compare with SKYNET and found that AOD errors introduce the largest variations. They also found that the sky irradiance calibration has a primary role  in the uncertainty of the retrieval, and they investigated influence of surface albedo and sphericity of aerosols, that was found negligible.

For the UVMFR data the uncertainty of the UVMFR SSA retrieval is mainly related to:

- direct to global irradiance measurements uncertainties.

- RTM input data accuracy.

Direct to global irradiance measurement uncertainties can result to a range of SSA values rather than a single value, that would produce a close match between the measurement and the RTM DGR outputs. This range broadens at low SZA and  high aerosol level cases, as shown in Figure 6, when the effect of the scattering/absorbing nature of aerosols in radiation is higher. The RTM inputs that were used for the SSA LUT construction include also an uncertainty budget (AOD, surface albedo, constant aerosol vertical profile, asymmetry factor). Following the uncertainty analysis of Krotkov et al. (2005b), the total relative uncertainty of the DGR measurement was calculated to be ±3%. AOD absolute uncertainty is considered as 0.02 for 368nm and 0.04 for 332nm, following the analysis of previous section. The impact of this on the SSA calculation is directly connected with AOD levels and the SZA. In figure 7 we have calculated the UVMFR SSA retrieval uncertainty for different AOD's and solar zenith angles, caused by DGR and AOD uncertainty. DGR and AOD uncertainty ranges from previous paragraph were used to calculate the possible SSA

range and the expected error. In the  figure, the mean AOD's , for each SZA bin (errorbars equal to one standard deviation)  for Athens measured by the UVMFR at each solar angle, are shown.

**3   SSA retrieval results**

Using the methodology described in the previous section we calculated the SSA at 332nm and 368nm using 1 minute data from the UVMFR. For the period under investigation, we also calculated the daily mean SSA's at these two wavelengths in the UV band and also the mean daily SSA's in the visible band derived from data provided by the CIMEL (L1.5 data) operating in Athens' AERONET station (figure 8).

[Figure]

**Figure 8** Daily mean  SSAs in the UV (UVMFR) at two wavelengths and at 440nm (CIMEL) for Athens area.

The variability of SSA during this period is quite high, ranging from 0.75 (0.62) to 0.98 (0.97) for  368nm (332nm) (2 standard deviations) with mean values of 0.90, 0.87 and 0.83 for 440nm, 368nm and 332nm respectively. In figure 9 we have calculated the mean monthly values of SSA at UV wavelengths and standard deviations for the whole period to examine the annual variability. The lowest SSA values were found for the period from February to May at both wavelengths, which should be linked to the usual dust events during this period for the  area, and also the presence of brown carbon. Paraskevopoulou, et al (2014), have found maximum values of Organic and Elemental Carbon, in February and November, in a 5 year (2008-2013) data set of in-situ measurements, at center of Athens. However, most months have similar SSAs, with

differences that lie well within the SSA variability of each month. Looking at the monthly mean AODs; despite the fact that standard deviations of both SSA and AOD's are large, it can be seen that higher AOD's are associated with less absorbing aerosol cases.

[Figure]

1     **Figure 9** Mean monthly SSAs (left axis) in the UV (UVMFR) at two wavelengths and AOD

2     at 368nm from the UVMFR (right axis) for the whole 5-year period, at Athens, errorbars at

3     represent one standard deviation of the mean.equal 1σ.

5     When calculating diurnal patterns of the SSA at UV and visible wavelengths for the Athens

6     area, we observed a mean diurnal pattern with a variability of the order of 0.02 to 0.05 and

7     having highest absorption (lowest SSA's) ±2 hours around noon (figure 10). Similar behavior

8     can also be seen from AERONET retrieved SSA's having higher values observed during the

9     early morning and late evening. However, the SZA limitation of the AERONET retrieval

10     methodology leads to lack of measurement points around noon. To investigate the uncertainty

11     in relation to UVMFR retrievals, the diurnal pattern was calculated for different SSA

12     uncertainty bins according to the analysis ofin the previous section. In general, the daily

13     pattern is clear for each bin and is mirrored by the AERONET inversion retrievals. However,

14     the statistical error one standard deviation1σ bars are quite large. These bars describeing the

15     variability of the SSA's during each hourly bin, are quite largbut also include the uncertainty

16     of the retrieved valueretrieval uncertainty.  We have to note that since no restriction has been

17     introduce for UVMFR SSA retrievals at low AOD's the uncertainty related with these data

18     becomes larger as seen also in figure 7.e.

[Figure]

21     **Figure 10** Diurnal patterns of SSA derived from the UVMFR and CIMEL measurements.

22     Mean values per hour plotted at 1σwith errorbars at one standard deviation. Local time in

23     Athens is UTC+2(winter) UTC+3(summer)

In order to investigate the possible dependence of SSA on AOD, figure 11 shows the synchronous UVMFR and CIMEL SSA retrievals plotted against AOD at 440nm. We found that in general, SSA decreases with a decrease in optical extinction, although lower AOD's are also linked to higher uncertainties of retrieved SSA. We believe that this behavior reflects seasonal changes in the average aerosol composition in Athens. Indeed, the annual cycle of SSA is the same as the AOD annual cycle having a maximum in summer and a minimum in winter. Studies of the SSA annual variability for other cities such as Ispra, Italy and Thessaloniki, Greece (Arola et al., 2005, Bais et al., 2005) revealed the same trend, with low SSA values (high absorption) associated with low AOD and reminiscent of mostly wintertime cases. It has to be noted that due to low AOD, uncertainties associated with the data obtained from both retrieval techniques (AERONET and UVMFR), are quite high. For higher AOD (>0.6), CIMEL retrievals show an almost constant value of the SSA ~0.92, while lower values have been retrieved at smaller AODs . Similar results were reported by Krotkov et al. (2005b) when analyzing measurements derived at at AERONET calibration site in Greenbelt, Maryland USA.

[Figure]

**Figure 11** Dependence of the calculated SSA from AOD measurements

We performed an analysis of the differences of SSAs between the visible and the UV parts of the spectrum based on aerosol characteristics using synchronous CIMEL and UVMFR SSA

1     retrievals and an aerosol classification scheme described in detail in Mielonen et al. (2009).

2     There, a classification of AERONET data was used in order to derive 6 aerosol types based

3     on the SSA measurement at 440nm and the AE that was derived in the 440-870 nm

4     wavelength range. Mielonen et al. (2009) used a visualization of this characterization, by

5     plotting AE versus SSA for individual sites, and compared their results with the CALIPSO

6     (Omar et al., 2005) aerosol classification scheme obtaining good agreement. In addition, the

7     difference between SSA at 440 nm and 1020 nm (similar to the approach applied by Derimian

8     et al. (2008)), was implemented to better distinguish fine absorbing aerosols from coarse ones.

9     The main idea was to fill this SSA versus AE aerosol type related "space" with the differences

10    of $SSA_{440}$-$SSA_{368}$ (SSADIFF) to investigate a possible link between SSA wavelength

11    dependence and aerosol type. In figure 12 using the Mielonen et al. (2009) aerosol typing

12    approach, we plot SSADIFF for different classes (colored scale), and separate aerosol types

13    by areas in the SSA/AE plot . In addition, actual points of $SSA_{440}$ retrieved by the CIMEL

14    instrument are shown in order to categorize Athens results according to the classification

15    scheme.

[Figure]

17    **Figure 12** Daily average $SSA_{440}$ (CIMEL) versus $AE_{(440-870nm)}$ . Colors represent different bins

18                   of the spectral differences of $SSA_{440nm}$- $SSA_{368nm}$.

20    The results  in figure 12 show that a mixture of aerosol types characterizes the ARSS site in

21    Athens, with $SSA_{440}$ values spanning all 6 sub-spaces. Analyzing the wavelength dependence

of the SSA, by defining SSADIFF as the difference $SSA_{440} - SSA_{368}$, there is evidence that high negative SSADIFF values (that means that the SSA at UV wavelengths is equal or relatively higher than $SSA_{440}$) tend to occur towards high AEs. For these cases (green color in figure 11) we observe high absorption cases with AE's around 1, which can be attributed  to polluted dust aerosol events. Also the majority of cases which comply with the condition AE < 0.7 are found with lower SSA at UV by at least 0.05 compared to $SSA_{440}$. More specifically, dust cases (mainly during spring) can be identified due to the proximity of Athens to the Saharan desert (Gerasopoulos, et al., 2010), explaining this behavior of absorbing aerosols at UV with low AE. Russell, et al., (2010) reported results obtained from diverse datasets showing SSA wavelength dependency from the IR down to visible wavelengths. In addition, Bergstrom et al. (2007) presented SSA spectra for dust-containing aerosols campaigns (PRIDE and ACE-Asia) including AERONET measurements at sites that are affected by dust such as Cape Verde, Bahrain (Persian Gulf) and the Solar Village (Saudi Arabia). Both studies concluded that the SSA spectra for AERONET locations, dominated by desert dust decrease with decreasing wavelength. In addition, Russel et al., (2010) reported that SSA spectra for AERONET locations dominated by urban-industrial and biomass-burning aerosols decrease with increasing wavelength in line with the results of Bergstrom et al. (2007). Figure 12 also shows that similar SSA values can be found for 440nm and 368nm and for fine aerosol cases (AE>1.4).

In order to understand the potential relative contributions of dust and brown carbon better, we applied the method of Schuster et al., (2016) to the AERONET measurements in Athens. This method separates contributions from black carbon, organic carbon, hematite and goethite,  to the retrieved refractive index at all available wavelengths, even in complex mixtures. Figure  13 shows the fractions of total aerosol volume attributed to these components, as well as the volume fractions accordingly. It is evident, according to this approach, that both brown carbon and mineral dust are likely absorbing components involved in the aerosol mixture in Athens, and brown carbon playing the more dominant role. Brown carbon highly absorbs in UV wavelengths and hardly any above 0.7nm (Kirchstetter et all, 2004). BrC fraction is higher  in October, but it has very large concentrations at the period March-June, which partly explains low SSA values at figure 9.

[Figure]

[Figure]

**Figure 13.** Total volume (in the upper plot) and volume fraction (in the lower plot) of absorbing aerosol components, as inferred by the method of Schuster et al. 2016. The retrieval gives the fractions for fine and coarse mode separately and here the contributions are shown as mode-weighted median value.

The utility of the AE for aerosol  extinction is that its value depends primarily on the size of the particles, ranging from a value of 4 for very small particles (Rayleigh scattering) to around 0 for very large particles (such as cloud drops). Thus AE for atmospheric aerosol  mixtures varies between limits specified by particle size. Various studies (e.g. Bergstrom et al., 2007) have used the Ångström Absorption Exponent (AAE) for studying the aerosol absorption wavelength dependence for different aerosol types and mixing (which is calculated similarly with the Ångström Exponent, only using Absorption Optical depth [AOD*(1-SSA)] instead of AOD). As the absorption AOD is a relatively smooth decreasing function with wavelength, it can be approximated with a power law wavelength dependence via the AAE which is defined as the negative of the slope of the absorption on a log-log plot.  14 Investigating the  temporal variability of $AAE_{(440-870)}$ and $AAE_{(332-440)}$ Measurements of $AAE_{(440-870)}$ are found to lie between 0.9 and 1.5 (2σ) in accordance with the results of Bergstrom et al., (2007). $AAE_{(332-440)}$ in the UV range is very different from that in the visible, with values ranging from 1.4 to 5 (2σ). A direct comparison reveals that for the aerosol composition features of Athens, the AAEs are usually up to 4 times higher in the UV range than in the visible. This is due to a combination of the enhanced absorption (lower SSA's) that has been found in the UV, together with higher AOD's in this band.

Finally, we have calculated mean CIMEL SSA values for all four retrieved wavelengths (440nm, 673nm, 870nm and 1020nm) for the whole period under study, and synchronous (5 minute SSA averaged around the CIMEL measurement time) UVMFR SSAs at UV (332nm and 368nm). The results are shown in figure 14 with errorbars at 1σ. Datasets of SSA retrievals are separated in 3 cases accordingly: a) all points (CIMEL L1.5 and all synchronous UVMFR data), b) measurements retrieved with AOD>0.2 (reduced uncertainty),  c) SSA retrievals for $AE_{340-440}$<0.7, to identify dust events and d) cases with AE>1.2 to include the fine mode cases. While for all cases the calculated standard deviation is quite high (≥0.05), there is a systematic SSA decrease in the UV range, and mean differences of 0.07 and 0.02 have been found when comparing SSA at the visible range and SSA at 332nm and 368nm respectively. Dust cases in particular show a spectral decrease in SSA with decreasing wavelength from 1022nm (CIMEL) down to 332nm (UVMFR). Fine mode cases show smaller spectral dependence ($SSA_{440nm}$-$SSA_{332nm}$ <0.03).

[Figure]

**Figure 14** Wavelength dependence of SSA from synchronous CIMEL and UVMFR measurements. Blue points represent all data points, red data retrievals with AOD>0.2, green points data with AE>1.2 and black data only dust aerosol cases. Vertical bars represent one standard deviation of the calculated mean.

The spectral dependence of the SSA from the visible to the UV wavelengths is in agreement with findings presented by Corr et al., (2009) and Krotkov et al., (2009). With Tthe same approach applied to Mexico City where measurements are also influenced by city emissions and blowing dust, Corr et al. (2009) studied the SSA behavior at UV wavelengths and showed that for AOD>0.1, SSA varied from 0.78 to 0.80 for 332nm and 368nm respectively with enhanced absorption at UV wavelengths relative to the visible wavelengths attributable to these types of aerosols. Krotkov et al., (2009) have modified a UVMFR in order to measure

also at 440nm , and found strong SSA wavelength dependence across blue and near UV spectral region.

**4 Conclusions**

Advantages of measuring the aerosol absorption (SSA) in the UV with the UVMFR instrument can be summarized as follows:

- AOD, in the UV wavelength range, is higher (for the same aerosol mass) than in the visible spectral range

- SSA retrievals with the uncertainty of ±0.03 can be derived for SZA> 40 degrees and with an uncertainty of ±0.04 for all SZAs where AOD>=0.2

- SSA retrievals are stable and repeatable over the five year period

We have analyzed a 5 year period of UVMFR and CIMEL measurements at the city of Athens retrieving SSA at visible and UV wavelengths based on the effect of aerosol SSA on the Direct to Global Ratio (DGR) for a given AOD and air mass. Since the CIMEL retrieval algorithm is more accurate for high SZA, the combination of the two instruments allows for an increase in measurement frequency of SSA and the ability to derive a complete diurnal cycle of aerosol absorption. In addition, the spectral differences of the aerosol absorption properties in the visible and UV wavelength range have been investigated, using synchronous CIMEL and UVMFR retrievals. Results of this work confirmed similar results found for Mexico City, Mexico (Corr et al., 2009), Washington DC, USA (Krotkov et al., 2005b) and Rome, Italy (Ialongo et al., 2010), that presented enhanced absorption of aerosols for UV wavelengths.

We have also used the produced dataset to investigate possible effects of aerosol type on observed SSA wavelength differences. The enhanced UV absorption can be mainly due to either dust or organic aerosol. Our analysis of Athens AERONET measurements suggests that the relative role of absorbing organic aerosol would be somewhat more significant than dust. The enhanced aerosol absorption found when comparing UV and visible spectrum results, shows that:

- We expect a systematic overestimation of modeled solar UV irradiance using SSA from extrapolation from the visible range as an input to RTMs

- There is a possibility  of a decrease in specific days/cases of regional $O_3$ due to the enhanced aerosol absorption (Li et al., 2005). But for the Athens case this could be verified only with Chemical model results.

- Satellite post-correction  results (e.g. Arola et al., 2009), including aerosol absorption effects, have to take into account absorption enhancement in the UV range.

- We expect an overestimation on the UV irradiance (UV Index) calculations on cloudless cases under dust and/or brown carbon presence when using SSA values from the visible range. This as a combination of the overestimated SSA and the high AODs during such events.

However, the spectral SSA differences, that we found, are well within the uncertainty of both retrievals as instrumental effects or absolute calibration uncertainties of sky radiances (~5% for the CIMEL almucantar measurements) might also play an important role when performing such comparisons. The coincidence of AOD measurements, from both instruments, using a single ETC for various SZA over the extended 5 year period used here, is a sign that no systematic SZA dependent factors influence the final SSA results.

The extended SSA dataset significantly improves comparative statistics and provides additional information on the effect of varying background aerosol conditions and higher aerosol absorption than that provided by Washington, DC, where dust aerosol cases are very rare. In conclusion, the combined use of CIMEL sun and sky radiance measurements in the visible with UVMFR total and diffuse irradiance measurements in the UV, provide an important advantage for remote measurements of column aerosol absorption over the UV-Visible spectral range.

**Acknowledgements**

P.Raptis would like to acknowledge the project «Aristotelis- SOLAR (50561), Investgation on the factors affecting the solar radiation field in Greece». V. Amiridis and S. Kazadzis would like to acknowledge the project "European Union's Horizon 2020 Research and Innovation Programme ACTRIS-2 (grant agreement no. 654109)"

---

## Author Comment (AC2) · 15 Nov 2016

We would like to thank the reviewer for the comments and recommendations. We have tried to include all the comments and corrections to the new manuscript.

*1,18: properties -> property ( Only one absorption property, i.e., SSA is retrieved).*

*1,19 5-years period*

*1,22: and study absorption spectral behavior of the [retrieved] SSA values*

*1,24: towards lower shorter UV wavelengths*

*1,25: High Strong SSA wavelength dependence*

All recommendations were included / corrected in the reviewed manuscript.

*1,27: "SSA decrease with decreasing extinction optical depth, suggesting an effect of the different aerosol composition" – this could be due to in increased SSA uncertainties at lower AODs*

A sentence was added in the abstract:
"However, part of this dependence, for low aerosol optical depths, is masked by the increased SSA retrieval uncertainty."

*2,2: "were investigated to understand seasonal variability of the results" – to explain?*

Changed in the reviewed manuscript.

*2.6. e.g.,*

Corrected

*2.7: IPCC references are missing*

Reference has been added.

*2,8: "as it appears that climate change is accelerating with aerosols impacting" - This sentence needs clarification and reference: how aerosol and climate changes are related?*

Sentence changed to:
However, a considerable amount of work still needs to be carried out as aerosols have an impact at local, regional and global climate scales.

*2,13: "significant aerosol absorption uncertainties in [modeled?] global Single Scattering Albedo (SSA),"*

Sentence was rephrased:

"Both emphasize that significant uncertainties in modeled Single Scattering Albedo (SSA) retrievals, constitute one of the largest single source of uncertainty in current modeling estimates of aerosol climate forcing."

*2,16:mixture [mixing state?]*

*2,20: 50% change [decrease?] in the [surface] erythermal irradiance*

**2,21: et al.,**

**2,22: e.g., (add comma)**

All changed in the reviewed manuscript.

*2,24 "a fixed irradiance path" – please reword. Surface irradiance is a result of averaging different direct and scattered photons arriving at the surface via different paths through the atmosphere.*

Sentence was re-written:

SSA calculated here differs from in situ SSA values retrieved from absorption and scattering measurements at a single altitude level (e.g., at the ground). Columnar SSA is a measurement arising from solar irradiance attenuation in the atmosphere.

*2,27: Do not use italic font in references.*

Changed in the reviewed manuscript.

*2,28: VIS-SSA -> column average SSA retrievals at the visible and near IR wavelengths (i.e., 440nm, 670nm, 870nm, 1020nm).*

Changed in the reviewed manuscript.

*2.29-2.32: "In addition, Goering et al. (2005), Taylor et al (2008) and Kudo et al. (2008) have proposed estimation techniques for the retrieval of spectral aerosol optical properties by combining multi-wavelength measurements using a priori constraints that are applied differently than in the single wavelength methods."*
*Suggest replacing this sentence with:*
*In addition, surface direct and diffuse irradiances had been used to derive spectral AOD and SSA at visible and UV wavelengths (King and Herman 1979; King 1979; Petters et al., 2003; Eck et al., 1998; Krotkov et al., 2005b; Bais et al., 2005; Goering et al., 2005; Taylor et al., 2008; Kudo et al., 2008; Corr et al., 2009).*

*2,32: "SSA retrieval in the ultraviolet (UV) part of the spectrum is weaker with large uncertainties." – I suggest removing this sentence.*

Sentence was replaced and references were added.
Sentence was removed.

*3,10: "like organic, nitrate and aromatic aerosols are still poorly known" add references, e.g., Jacobson, M. Z. (1999), Isolating nitrated and aromatic aerosols and nitrated aromatic gases as sources of ultraviolet light absorption, J. Geophys. Res., 104, 3527–3542.*

Reference added

*3,16-18: Barnard et al. (2008) [ and Corr et al., (2009) ]in a case [field] study found that, in the near-UV spectral range (250 300 to 400 nm) 3,28: "in at 870 [nm] could be a hint reason for.."*

Sentences corrected as recommended.

*" 3,32: " using Brewer [direct sun and global irradiance spectral UV ] measurements*

The sentence has been restated.

*4,2: and They used imaginary refractive index and found*

Changed in the reviewed manuscript.

*4,10: (e.g. Zerefos et al., 2012;) - add more references*

References added:
Reuder, J., and H. Schwander (1999), Aerosol effects on UV radiation in nonurban regions, J. Geophys. Res., 104(D4), 4065–4077, doi:10.1029/1998JD200072.

Krzyścin, J. W., and S. Puchalski (1998), Aerosol impact on the surface UV radiation from the ground-based measurements taken at Belsk, Poland, 1980–1996, J. Geophys. Res., 103(D13), 16175–16181, doi:10.1029/98JD00899.

Balis, D. S., Amiridis, V., Zerefos, C., Kazantzidis, A., Kazadzis, S., Bais, A. F., Meleti, C., Gerasopoulos, E., Papayannis, A., Matthias, V., Dier, H., and Andreae, M. O.: Study of the effect of different type of aerosols on UV-B radiation from measurements during EARLINET, Atmos. Chem. Phys., 4, 307-321, doi:10.5194/acp-4-307-2004, 2004.

*4,11: "comparable in magnitude [or exceeding ] with those caused by the decline in stratospheric ozone [depending on wavelength]*

Changed in reviewed manuscript.

*4,13"reduction of 7% of AOD  - at what wavelength?*

This trend was calculated at 305nm, and this info has been added to the manuscript.

*4,17:  "tropospheric photochemistry[, causing:]*

*5,5: "Solar irradiance satellite retrieval algorithms are directly affected " -> satellite retrieval algorithms of the surface UV irradiance are directly affected*

*5,8: absent from[current] satellite (e.g., OMI) retrieval algorithms*

*5,12: "Uncertainty on [in] commonly used .."*

*5,13: "fall short in precision due to large uncertainties in the input parameters" -> The model accuracy of the surface UV irradiance is limited by large uncertainties in the input parameters*

The sentences has been restated/corrected.

**5,24- 6,10 – suggest deleting this paragraph as common knowledge.  Move Equation (2) to the beginning of section 2.2**

Deleted as suggested. Equation moved to 2.2.

*6,11-27 – this paragraph looks repetitive and could be blended with the earlier part of introduction.*

Corrected  as suggested

*7,25: constructing [manufacturing] company*

*7,29: irradiance[s]*

*8,5: "in conjunction with radiative transfer model (RTM) calculations that have been performed using the Libradtran code (Mayer and Kylling, 2005). "*

Sentences corrected

*8,12: "SSA is a key aerosol optical property and describes the portion of solar irradiance that is scattered from the main direct beam passing through the atmosphere." – Equation (2) can be removed after this sentence.*

Corrected as suggested

*9,4:  "raw voltage measurements [corrected for night-time voltages and non-ideal angular response] could be used."*

*9,19: "dt." -> time interval*

*10,fig 2 caption "for a day with variable cloudiness [in the afternoon]*

*10, 9 : "for performing [determining] extraterrestrial Langley calibration constant (ETC) determination*

*10,10: "the Beer-Lambert law for to the UVMFR direct [voltage] measurements"*

All sentences corrected

**10,13-14: "extrapolated AOD at UVMFR wavelengths" – Clarify how AOD was extrapolated?**

In the next paragraph we explain in detail how the extrapolation was performed, using least square quadratic spectral extrapolation.

**11,6: AOD's at 332 nm and 368 were**

Corrected

**11, 15: extrapolation [using ln(AOD)versus ln(wavelength) ]?**

The sentence has been restated to provide info on specific wavelengths used in the extrapolation process. "..we first calculated the CIMEL derived AOD at 332 nm and 368 nm, applying least square quadratic spectral extrapolation, using ln(AOD) as function of ln(wavelength) from AERONET measurements at 340nm 380nm, 440nm and 500nm."

**12, Fig 4 caption: "Comparison of CIMEL and[extrapolated] and UVMFR retrieved AODs for..**

**332 nm (up top panel) and 368 nm (down bottom panel)."**

**12,9: "as a function of SZA (figure 5 4)."**

**12,12: "are included [found]**

Corrected.

**12,18: "due to instrumental (filter related) changes" – Most likely reason for ETC change is UVMFR Teflon diffusor contamination. Explain how often the UVMFR diffusor was cleaned and what cleaning procedures applied?**

We have added this information:
"The instrument's Teflon diffuser contamination is the most common reason for long term changes in the ETC. Maintenance procedure for the Athens instrument included cleaning and inspection of the diffuser and check of the levelling and shadowing, three times a week. In addition, metal spikes have been built around the instrument to avoid the diffuser destruction by birds. "

**12, 19 AODs 's deviations errors on in SSA calculations ..**

**13,6: were deployed for the use of [used for construction of] the LUT**

**14, Fig 6. Caption Figure 6 LUT of direct to global ratio at 368nm color bar represents assumed SSA values.**

*14,13: average annual monthly AODs 's*

*15,12: SSAs 's*

*15,15: role on [in] the uncertainty*

*15,21: a [close] match between*

*15,22: "This range broadens at low SZA and low aerosol level cases" – Please, clarify this sentence and refer to Figs 5 and 6.*

The sentence has been restated.

*15, 26: "AOD uncertainty is considered as 2% for 368nm and 4% for 332nm," Should it be absolute AOD uncertainties: 0.02 at 368nm and 0.04 at 332nm ?*

*16,3 AODs*

*16,5: In the same figure,*

*16,6: mean AODs 's [for each SZA bin] and 1, the error bars equal to one standard deviation are shown*

*16,16: Figure 8 Daily Mean daily SSAs*

*16,20: for 332nm (368nm)" - Should it be reversed, i.e. , at 368nm (332nm) (SSA at 332 nm is generally lower than at 368nm) – fig 7. ?*

*17, Fig. 9 : Include X axis title.*

All above recommendations from 12,19 to 17 have been taken into account.

*Are spectral differences between SSA at 368nm and at 332nm between November and March statistically significant? Apply standard statistical significance tests*

We have tried to investigate this issue. We have applied the student t-test and we have found the differences significant on a 99% level.

The standard deviations of the monthly SSA shown in figure 9 are ranging from 0.05 to 0.1 (6% to 12% of the mean at the specific months). But the standard deviations of the differences are in the range of 0.01, to 0.018, while mean differences per month range from 0.022 (July) to 0.074 (November). In addition data points are in the range of 20.000 to 100.000 per month.

So that means that even if calculated SSAs for each wavelength vary for a specific month, the spectral difference variability is lower. Here is the mean differences per month and the red limits represent the area that 95% of the data are included.

[Figure]

*17,1: for the specific area," – correct reference Pareskevopoulouet al (2014) ->*
*Paraskevopoulou, et al.,*

*17,2: at in February and November*

*17, 3 at in a 5 year (2008-2013)*

*17,4:  have similar behavior SSAs*

*17,16-17:  "for different SSA uncertainty bins according to analysis of in the previous*
*section*

Recommendations have been taken into account

**18, Figure 10: Suggest local time or SZA as X axis**

[Figure]

The Figure 10 main purpose was to reveal the diurnal cycle of SSA. Lowest values around noon are observed year round, independently of the SZA (see attached figure), indicating a change to more absorbing aerosols at that time. This behavior is recorded by both instruments. Although CIMEL does not have retrievals around noon. We have plotted SSA as a function of solar zenith angle as recommended and in order to assess that the results are angle independent. This plot is not included in the manuscript, as it is not adding any additional information.

*18, Figure 10 caption: Mean values per hour plotted [with error bars showing one standard deviation] at 1*

*18,8: are [also] linked*

*18,8: derived at AERONET calibration site in Greenbelt, Maryland USA Washington*

*19,15: link between [SSA] wavelength dependence and*

*20, 5 The results of in figure 12*

*20, 9 relatively higher than SSA440) tend [to occur] towards high AEs*

*20,10 attributed in [to] polluted*

*21,6: Schuster et al., (2016)*

*21,7: method separates [contributions from] black carbon, organic carbon, hematite and goethite, using [to the retrieved] refractive index*

Recommendations have been taken into account

*21,9: 8-9: "Figure 12 shows the fractions of total aerosol [column] volume attributed to these components in both fine and coarse mode. This should be Fig. 13. There are no fine and coarse mode fraction data in Fig.13.*

We have included the original version of figure 13.

[Figure]

[Figure]

**21,14: " higher at [in October] OCTOBER " – It will be interesting to explain BrC peak in October compared to other months. Are there in-situ measurements in Athens that could support this finding?**

Paraskevopoulou, et al (2014), have studied a 5 year data set of in situ measurements in Athens and have found maximum values of Organic and Elemental Carbon, in February and November. We refer to that in the discussion of figure 9. We do not have any additional measurements to validate this result. The figure comes from the Schuster et al., 2016 publication.

**21,17: Figure 13 caption: "Volume fraction[s] (in the lower plot) of absorbing aerosol components**
Corrected

"

**22,4: " for atmospheric aerosol [mixtures] scattering varies**

**22,10-11 "Figure 13 shows the temporal variability of AAE(440-870) and AAE(332-440)." – This figure (14?) is missing**

Recommendations have been taken into account

*23, 7. I suggest adding new reference, which shows previously measured AERONET-UVMFR SSA spectral dependence in Thessaloniki, Greece: N. Krotkov ; G. Labow ; J. Herman ; J. Slusser ; R. Tree ; G. Janson ; B. Durham ; T. Eck ; B. Holben; Aerosol column absorption measurements using co-located UV-MFRSR and AERONET CIMEL instruments. Proc. SPIE 7462, Ultraviolet and Visible Ground- and Space-based Measurements, Trace Gases, Aerosols and Effects VI, 746205 (August 20, 2009); doi:10.1117/12.826880.*

The reference has been added.

*23,19: for all SZA[s]*

*24,10: "We have also [used] the produced dataset to investigate*

Recommendations have been taken into account

*24,18: "We expect a possible decrease in specific days/cases of regional O3 due to the enhanced aerosol absorption" - This conclusion is not supported in the main text. Add Chemical model results to support this.*

We have changed the sentence to:

There is a possibility of a decrease in specific days/cases of regional $O_3$ due to the enhanced aerosol absorption (Li et al., 2005). But for the Athens case this could be verified only with Chemical model results.

**Aerosol absorption retrieval at ultraviolet wavelengths in a complex environment**

**S. Kazadzis[1,2], P.I. Raptis[2], N. Kouremeti[1], V. Amiridis[3], A. Arola[4], E. Gerasopoulos[2], G.L. Schuster[5]**

[1]{Physikalisch-Meteorologisches Observatorium Davos, World Radiation Center (PMOD/WRC) Dorfstrasse 33, CH-7260 Davos Dorf, Switzerland}

[2] {Institute of Environmental Research and Sustainable Development, National Observatory of Athens, Greece}

[3]{Institute of Astronomy Astrophysics, Space Applications and Remote Sensing, National Observatory of Athens, Greece}

[4] {Finnish Meteorological Institute, Kuopio Unit, Finland}

[5] {NASA Langley Research Center, Hampton, VA, USA}

Correspondence to: stelios.kazadzis@pmodwrc.ch

**Abstract**

We have used total and diffuse UV irradiance measurements with a multi-filter rotating shadow-band radiometer (UVMFR), in order to calculate aerosol absorption  property (Single Scattering Albedo - SSA) in the UV range, for a  5-years period in Athens, Greece. This data set was used as input to a radiative transfer model and the SSA for 368nm and 332nm has been calculated. Retrievals from a collocated CIMEL sun-photometer were used to validate the products and study absorption spectral behavior of retrieved SSA values at these wavelengths. UVMFR SSA together with synchronous,CIMEL-derived, retrievals at 440nm, show a mean of 0.90, 0.87 and 0.83, with lowest values (higher absorption) towards  shorter wavelengths. In addition, noticeable diurnal variations of the SSA in all wavelengths are revealed, with amplitudes in up to 0.05.  Strong SSA wavelength dependence is found for cases of low Ångström exponents and also an SSA decrease with decreasing extinction optical depth, suggesting an effect of the different aerosol composition.

However, part of this dependence, for low aerosol optical depths, is masked by the increased SSA retrieval uncertainty. Dust and Brown Carbon UV absorbing properties were investigated to  explain seasonal variability of the results.

**1   Introduction**

The role of aerosols, both natural and anthropogenic, is extremely important for regional and global climate change studies as well as for overall pollution mitigation strategies (e.g., IPCC, 2013). However, a considerable amount of work still needs to be carried out as aerosols have an impact at local, regional and global climate scales. Furthermore, the components controlling aerosol forcing, account for the largest uncertainties in relation to anthropogenic climate change  IPCC, 2034). A comprehensive review of the assessment of the aerosol direct effect, its state of play as well as outstanding issues, is given by *(IPCC, 2034)* and *(Yu et al., 2006)*. Both emphasize that  significant  uncertainties in modeled  Single Scattering Albedo (SSA) retrievals, constitute one of the largest single source of uncertainty in current modeling estimates of aerosol climate forcing. SSA is the ratio of scattering to total extinction (scattering plus absorption), and it depends strongly on chemical composition, particle size, mixing state, relative humidity and wavelength. Comprehensive measurements are crucial to understand their effects and to reduce SSA uncertainties that propagate into aerosol radiative forcing estimates. For example for the same aerosol load (aerosol optical depth), the absorbing nature of aerosols can lead to up to 50%  decrease in the erythermal irradiance, compared to only scattering aerosols (Bais et al., 2014). SSA calculated here differs from in situ SSA values retrieved from absorption and scattering measurements at a single altitude level (e.g., at the ground). Columnar SSA is a  measurement arising from solar irradiance attenuationtransfer in the atmosphere.

In the visible (VIS) and in the near infrared (NIR) parts of the spectrum, advanced retrieval algorithms for microphysical aerosol properties have been developed in the framework of the Aerosol Robotic Network (AERONET) and the Skyradiometer Network (SKYNET) (e.g., Dubovik and King, 2000; Nakajima et al., 1996). All AERONET stations currently provide inversion based  column average SSA retrievals at the visible and near IR wavelengths (i.e., 440nm, 670nm, 870nm, 1020nm)VIS SSA retrievals. In addition, surface direct and diffuse irradiances had been used to derive spectral AOD and SSA at visible and UV wavelengths (King and Herman 1979; King 1979; Petters et al., 2003; Eck et al., 1998; Krotkov et al., 2005b; Bais et al., 2005; Goering et al., 2005; Taylor et al., 2008; Kudo et al., 2008; Corr et al., 2009). In addition, Goering et al. (2005), Taylor et al (2008) and Kudo et al. (2008) have proposed estimation techniques for the retrieval of spectral aerosol optical properties by combining multi-wavelength measurements using a priori constraints that are applied differently than in the single wavelength methods. SSA retrieval in the ultraviolet (UV) part of the spectrum is weaker withhas  large uncertainties and is rarely available. As AERONET does not provide any information about SSA at the UV, compared to the visible- and infrared spectral region, only a few publications have dealt with aerosol absorption at UV wavelengths (e.g. Eck et al., 1998; Krotkov et al., 2005a; Bais et al., 2005; Corr et al., 2009). It is envisaged that improvement in measurement precision accuracy and in the general understanding of aerosol absorption in the UV (and immediate derivatives like the SSA) in various scientific applications, will contribute significantly to enhancing the accuracy of UV related radiation forcing estimates. For example, desert dust particles (Alfaro et al., 2004), soot produced by fossil fuel burning, and urban transportation, all strongly absorb UV radiation. However, optical properties of other potential UV absorbers like organic, nitrate and aromatic aerosols are still poorly known (Jackbson, 1999). Torres et al., (2007), in an overview study of OMI aerosols products, summarized the algorithmical techniques of SSA satellite retrieval at 388nm, which uses spectral variability between 354nm  and 388nm , 388nm reflectance and a selection on the aerosol type. They compared to AERONET SSA at 440nm and found a root mean square error of 0.03.  Bergstrom et al., 2003 showed that spectra of aerosol SSA obtained in different campaigns around the world differed significantly from region to region, but in ways that could be ascribed to regional aerosol composition. Moreover, results from diverse air, ground, and laboratory studies, using both radiometric and in situ techniques, show that the fractions of black carbon, organic matter, and mineral dust in atmospheric aerosols play a role in the determination of the wavelength dependence of aerosol absorption (Russell et al., 2010). Barnard et al. (2008) and Corr et al., (2009), investigating the variability of SSA in a case field study for the Mexico City metropolitan area, found that, in the near-UV spectral range (250 300 to 400 nm), SSA is much lower compared to SSA at 500 nm indicative of enhanced absorption in the near-UV

range. They suggested that absorption by elemental carbon, dust or gas alone could not account for this enhanced absorption leaving the organic carbon component of the aerosol as the most likely absorber. It has been found in many studies that, in addition to dust, the absorbing organic carbon compounds can induce strong spectral absorption increasing towards the shortest UV wavelengths. Sources of these light–absorbing organic carbon compounds (often called as "Brown Carbon", BrC) are various; biomass burning (e.g. Kirchstetter et al.2004), urban smoke (e.g. Liu et al. 2015) and biogenic emissions (e.g. Flores et al. 2014).

Corr et al. (2009) presented a review of studies estimating SSA at different wavelengths. For the visible part of the spectrum, two different approaches have been presented. The first (Dubovik et al., 2002), introduced sky radiance measurements in a matrix inversion technique to calculate various aerosol microphysical properties. This methodology has been widely applied in the AERONET. The second (Eck et al.,2003, (Kassianov et al., 2005), proposed the use of radiative transfer model (RTM) calculations, using as input measurements of AOD and the ratio of direct to diffuse irradiance at specific wavelengths. However, in the case of SSA calculations at UV wavelengths, enhanced measurement uncertainties, RTM input assumptions, and interference of absorption by other gases ($O_3$, $NO_2$), make the retrieval more difficult. All reported results concerning UV-SSA, utilize RTM combined with total and diffuse relative irradiance measurements (Herman et al., 1975; King and Herman 1979; King 1979; Petters et al., 2003; Krotkov et al., 2005b; Corr et al., 2009; Bais et al., 2005) or absolute irradiance measurements (Kazadzis et al., 2010; Ialongo et al., 2010; Bais et al., 2005). The review made by Corr et al. (2009) also presents the major differences in the results of simulations of the SSA, arising from RTM input assumptions, measurement techniques and retrieved wavelengths. An additional problem is that previous studies have dealt with short time periods due to the limited lifespan of experimental campaigns.

Moosmuller et al (2012) showed that iron concentration in mineral dust aerosols is linked to lower SSA at 405nm than in 870, which could be a hint reason for lowest SSA in the UV-VIS range during dust events. Medina et al (2012) found in El Paso-Juarez also large variation in UV range SSA, with lower values than visible wavelengths and showed that on heavy polluted days it can get as low as 0.53 at 368nm. An effort was made to calculate SSA in lower UV wavelengths, using Brewer (Direct and Global Spectral Irradiance at UV range)

measurements, at Belgium, revealing lowest values but with high uncertainty (Nikitidou et al, 2013). Recently Schuster et al (2016) have tried to distinguish aerosol types, by their optical properties and assumed that dust particles have higher absorption at UV wavelengths, and. They used imaginary refractive index spectral dependence to separate from black carbon and infer hematite/goethite in the coarse mode. They found that dust particles containing hematite are highly absorbing in the UV region.

Ultraviolet (UV) solar radiation has a broad range of effects on life on Earth (UNEP et al., 1998;UNEP et al., 2007;UNEP, 2003). It influences not only human beings (e.g. (Diffey, 1991)), but also plants and animals (e.g.  Bornman and Teramura, 1993). Furthermore, it causes degradation of materials and functions as a driver of atmospheric chemistry. There are various studies linking changes of the UV radiation field with changes in the scattering and absorption of aerosols in the atmosphere (e.g.  Zerefos et al., 2012, Balis et al., 2004, Reuder and Schwander, 1999, Krzyścin and S. Puchalski 1998). Such changes can be comparable in magnitude to 
[revised manuscript text omitted]

[Figure]

**Figure 1.** UVMFR angular response function at 368nm channel, normalized to the ideal (cosine) angular response. 2 sets of responses one from the south to north scan and one from the west to east are presented.

**2.2 Retrieval methodology**

SSA is a key aerosol optical property and describes the portion of solar irradiance that is scattered from the main direct beam passing through the atmosphere. Changes in SSA influence mostly the diffuse radiation reaching the earth's surface, while its effect on direct radiation can be considered negligible.  SSA at a wavelength λ provides the contribution of aerosol particle scattering relative to the total extinction (absorption plus scattering),

$$SSA = \frac{b_{sca}(\lambda)}{b_{abs}(\lambda) + b_{sca}(\lambda)} \tag{1}$$

Theoretically  SSA values range from 0 (totally absorbing aerosols ) to 1 (totally scattering aerosol). Actual SSA values in the atmosphere can be found usually in the range of , although it cannot be less than 0.2 because of the light diffracting through the aerosols, and in the atmosphere is usually in the range 0.65 0.5 to 1 (Corr et al., 2009). The asymmetry parameter, is the phase function (P) weighted average of the cosine of the scattering angle (θ) over all directions. Assuming azimuthal symmetry, the scattering angle integration extends from – π to +π such that the asymmetry parameter (g) is given by

$$g = \frac{1}{2} \cdot \int_{-\pi}^{\pi} cos\theta \cdot P(\theta) \cdot sin\theta \cdot d\theta \qquad\qquad (2)$$

Values for g range from -1 (backscattered radiation only) to 1 (forward scattered radiation only) in theory, and from 0 to 1 for particles in the atmosphere.

Model calculations can be used for retrieving SSA when global and/or diffuse spectral irradiance, solar zenith angle (SZA), total column ozone, and AOD are known (Krotkov et al., 2005b; Kazadzis et al., 2010; Ialongo et al., 2010; Corr et al., 2009; Bais et al., 2005). In our retrieval methodology we have used partly the basic approach that is described in detail in the Corr et al. (2009), Krotkov et al. (2005a) and Krotkov et al. (2005b). This approach consists of measurements of the direct to global irradiance ratios (DGR) and AODs measured with the UVMFR instrument for our case, that are used as basic input parameters to the RTM for the calculation of the SSA at 332nm, and 368nm. These wavelengths are selected for having the lowest ozone absorption from the seven available (Bass and Paur ,1985). The advantage of this method is that the same detector and filter measure global and direct irradiance, thus there is no need for absolute irradiance calibration and raw voltage measurements --corrected for nighttime voltages and angular response - could be used.

Global irradiance measurements from the UVMFR have been used in order to distinguish cloud free conditions for each of the one minute measurements. Clouds are detectable in the measured UVMFR global irradiance (GI) (at 368nm) since they cause larger variability than aerosols. For distinguishing between cloudy and cloud free conditions, we have applied an updated version of the method of Gröbner et al., (2001). The method is based on the comparison of the measured global irradiance with radiative transfer calculations for cloud free conditions and quality assurance is checked by the following criteria:

a. The measured GI has to lie within the modeled (cloud free) GI for a range of aerosol loads (AOD at 500 nm of 0.1 and 0.8, respectively), corresponding to the 5th and 95th percentile of the AERONET data for the examined location and period b. The rate of change in the measured GI with SZA has to be within the limits depicted by the modeled cloud free GI, otherwise the measurements are assumed cloud contamination.

c. All measured GI values within a time window (dt= ±10 min) should be within 5% of the modeled cloud free GI, and adjusted to the level of the measurement, using an integral over time interval.

If at least 85% of the points in dt pass tests a) – c), then the central point is flagged as cloud free. In this study, we have allowed a tolerance level of ±10% for tests a) and b) in order to compensate for differences between the modeled GI and measured GI due to instrumental uncertainties, as well as for usage of average climatological parameters (constant total ozone column, SSA, e.t.c.) as inputs to the model.  We have limited the method to SZA<70º to avoid uncertainties related with low solar irradiance levels. An example of the results of the method is presented in figure 2 for a day with variable cloudiness. It has to be noted that in all

CIMEL-UVMFR comparisons, using synchronous measurements, both the above method and

AERONET cloud  screening algorithm  (presented by Smirnov, et al, 2000) are taken into account.

[Figure]

**Figure 2**. Determination of cloudless 1-minute measurements (red), from all measurements (blue) for a day with variable cloudiness in the afternoon.

Measurements of the diffuse and global irradiance from the UVMFR have been used in order
to retrieve the direct irradiance at 332nm and 368nm. We used the AERONET database to
select days with very low AOD (<0.1). For the urban environment of Athens such cases are
related with the presence of northern winds. Afterwards we selected cloudless sky half-days
for  determining extraterrestrial Langley calibration constant (ETC)
by applying the Beer-Lambert law  UVMFR direct voltage measurements. $V_{0langley}$ in
figure 3 represent the half day values calculated with this method. In order to examine the
consistency of this approach we calculated the $V_{0cimel}$ also as

$$V_{0\,cimel} = V e^{\mu\,(AOD_{cimel} + \tau_{rayleigh})}$$

where V is the voltage measured by UVMFR, $\mu$ is the air mass, $AOD_{cimel}$ is the extrapolated
AOD at UVMFR wavelengths and $\tau_{rayleigh}$ is Rayleigh scattering optical depth. Daily averages
of $V_{0cimel}$ for the selected days were compared with $V_{0langley}$ as presented at figure 3. These
independent approaches appear stable through the years, with no obvious drift or change, so
we decided to use a single ETC for the whole period for each wavelength.

[Figure]

**Figure 3**. ETC values at 368nm, calculated using Langley plots of UVMFR measurements,
and Using Cimel extrapolated AOD's as input, for selected (low AOD's and clear sky) days
for the whole period

AOD's at 332 nm and 368 were calculated using the selected UVMFR derived ETC. In
contrast with the Krotkov et al., 2005a approach we have not transfered the CIMEL ETCs to the UVMFR measurements; rather, we have independently calculated UVMFR-based AODs.

Validation of the results was performed based on synchronous UVMFR and CIMEL

measurements. The mean AOD calculated from the 1 minute UVMFR measurements within

±5 minutes from the CIMEL measurement (when the UVMFR 10 minute period is characterized by cloudless conditions) has been defined as synchronous. Since the CIMEL

instrument provides measurements of AOD at 340 nm and 380 nm, we first calculated the

CIMEL derived AOD at 332 nm and 368 nm,  applying least square quadratic spectral extrapolation, using ln(AOD) as function of ln(wavelength) from AERONET

measurements at 340nm 380nm, 440nm and 500nm.  (Eck et al, 1999).

[Figure]

**Figure 4**. Comparison of CIMEL and UVMFR retrieved AODs for synchronous measurements for 332 nm (left panel) and 368 nm (right panel).

The results of this comparison have a Pearson product moment correlation coefficient equal to 0.96 and 0.98 respectively for 332nm and 368nm AODs. Mean differences were zero, with standard deviations of 0.031 and 0.025 for the respective wavelengths, comparable with the CIMEL AOD retrieval uncertainty of ±0.02. The quality of the data produced can be verified by comparing the AOD's retrieved by the two instruments as a function of SZA (figure 545). Relative stability of the AOD differences (that are in the order of the AERONET uncertainties), verifies the validity of the calibration of the UVMFR AOD's. and the fact that no SZA dependent errors (that would be directly related with an erroneous ETC determination) are included found in this procedure. An AOD, SZA dependent trend, in the order of 0.02 (if excluding the 15$^{\circ}$ SZA bin) can be observed which could be attributed to ETC determination uncertainty or non ideal correction for the cosine response error of the UVMFR.

In figure 5, AOD's have been grouped in bins of 5 degrees (of SZA). The differences shown in figure 5 include ETC determination accuracy, the extrapolation of CIMEL AOD at 368nm, together with instrumental/measurement errors. Using a single UVMFR ETC for the whole period provides very good agreement between the two instruments. However, this may not be the case for all UVMFR instruments using this approach as ETC may suddenly or gradually change especially for years-long time series due to instrumental (filter related) changes. AOD's deviations could lead to large errors on in SSA calculations, so this comparison ensures that these errors are minimized. The instrument's teflon diffuser contamination is the most common reason for long term changes in the ETC. Maintenance procedure for the Athens instrument included cleaning and inspection of the diffuser and check of the levelling and shadowing, three times a week. In addition, metal spikes have been built around the instrument to avoid the diffuser destruction by birds.

[Figure]

**Figure 5** AOD differences between CIMEL and UVMFR at 368 nm, as a function of solar

                                 zenith angle.

We calculated look up tables (LUT) with the RTM, of DGR at 368nm and 332nm as a function of SZA, AOD, SSA, asymmetry factor (g) and total column ozone.

CIMEL/AERONET mean daily ozone values and climatological –satellite derived  NO$_2$

values were used for construction of the LUT while for g, we used the mean daily value as retrieved at 440nm from the CIMEL instrument measurements when available and the mean value of the whole period equal to 0.7 ($2\sigma$ standard deviation of the g during this period was 0.04) otherwise. Using UVMFR AOD and DGR measurements, we then calculated the matching SSA values for each individual UVMFR DGR measurement.

LUT examples are visualized in figure 6, for clarification of the method. For known SZA and

AOD (in cloudless sky conditions), the variability of the DGR is caused by aerosol properties other than AOD. At low aerosol loads this variation is nearly negligible, but it becomes more important at higher aerosol load. More absorbing aerosols lead to smaller values of DGR.  It is crucial to observe the range of SSAs in the two examples. For low AOD's, accurate SSA

determination requires very low uncertainty of the DGR and the AOD measurement. While for high AOD's the range of DGRs for a particular SZA is quite large.

[Figure]

**Figure 6** LUT of direct to global ratio at 368nm, as calculated for AOD 0.1 (left) and 0.8 (right) with respect to SZA (g=0.7), colourbar represents assumed SSA values.

**2.3 Retrieval Uncertainties**

The CIMEL sunphotometer provides SSA inversion retrievals characterized as Level 1.5 and Level 2.0 data. Level 2.0 (L2) data are recommended by AERONET as they have less uncertainty but are restricted in measurement to SZA>50 degrees, AOD at 440 nm> 0.4 and homogeneous sky conditions. These limitations make AERONET SSA L2 worldwide measurements unsuitable for:

a. climatological studies due to the AOD restriction that limits analyses only to areas having large average annual AOD's, or to cases of moderate to high aerosol episodes in specific areas. As an example, for the urban site of Athens, which is one of the most polluted cities in Europe, the number of measurements is limited to an average 11 cases per month for the whole analysis period.

b. diurnal variation studies due to the SZA restriction. For mid and low latitude sites, this limitation leads to a severe lack of information on diurnal SSA patterns as there are only few wintertime measurements and close to zero measurements at local noon.

[Figure]

**Figure 7** Uncertainty estimate (color) for the SSA retrieval from UVMFR as a function of AOD and solar zenith angle for 368nm (left panel) and 332nm (right panel), based on DGR and AOD uncertainties. Superimposed, mean AODs for 2.5 degree bins of solar zenith angle are shown.

Level 1.5 data: AERONET Level 1.5 (L1.5) SSA data are provided by AERONET for all AOD's and at all SZA that almucantar scans are performed. In this work L1.5 data were used, but with an extra quality control. We have ignored SSA L1.5 data when L2 size distribution is not available. Thus we have an enhanced L1.5 SSA data set with AOD<0.4, but with L2 cloud screening, calibrations and quality controls. Data has been compared with UVMFR retrieved SSA's taking into account limitations related with the retrieval uncertainties. Khatri et al (2016) studied AERONET SSA retrieval uncertainties, in order to compare with SKYNET and found that AOD errors introduce the largest variations. They also found that the sky irradiance calibration has a primary role  in the uncertainty of the retrieval, and they investigated influence of surface albedo and sphericity of aerosols, that was found negligible.

For the UVMFR data the uncertainty of the UVMFR SSA retrieval is mainly related to:

- direct to global irradiance measurements uncertainties.

- RTM input data accuracy.

Direct to global irradiance measurement uncertainties can result to a range of SSA values rather than a single value, that would produce a close match between the measurement and the RTM DGR outputs. This range broadens at low SZA and  high aerosol level cases, as shown in Figure 6, when the effect of the scattering/absorbing nature of aerosols in radiation is higher . The RTM inputs that were used for the SSA LUT construction include also an uncertainty budget (AOD, surface albedo, constant aerosol vertical profile, asymmetry factor). Following the uncertainty analysis of Krotkov et al. (2005b), the total relative uncertainty of the DGR measurement was calculated to be ±3%. AOD absolute uncertainty is considered as 0.02 for 368nm and 0.04 for 332nm, following the analysis of previous section. The impact of this on the SSA calculation is directly connected with AOD levels and the SZA. In figure 7 we have calculated the UVMFR SSA retrieval uncertainty for different AOD's and solar zenith angles, caused by DGR and AOD uncertainty. DGR and AOD uncertainty ranges from previous paragraph were used to calculate the possible SSA

range and the expected error. In the  figure, the mean AOD's , for each SZA bin (errorbars equal to one standard deviation)  for Athens measured by the UVMFR at each solar angle, are shown.

**3    SSA retrieval results**

Using the methodology described in the previous section we calculated the SSA at 332nm and 368nm using 1 minute data from the UVMFR. For the period under investigation, we also calculated the daily mean SSA's at these two wavelengths in the UV band and also the mean daily SSA's in the visible band derived from data provided by the CIMEL (L1.5 data) operating in Athens' AERONET  station (figure 8).

[Figure]

**Figure 8** Daily mean  SSAs in the UV (UVMFR) at two wavelengths and at 440nm (CIMEL) for Athens area.

The variability of SSA during this period is quite high, ranging from 0.75 (0.62) to 0.98 (0.97) for  368nm (332nm) (2 standard deviations) with mean values of 0.90, 0.87 and 0.83 for 440nm, 368nm and 332nm respectively. In figure 9 we have calculated the mean monthly values of SSA at UV wavelengths and standard deviations for the whole period to examine the annual variability. The lowest SSA values were found for the period from February to May at both wavelengths, which should be linked to the usual dust events during this period for the  area, and also the presence of brown carbon. Paraskevopoulou, et al (2014), have found maximum values of Organic and Elemental Carbon, in February and November, in a 5 year (2008-2013) data set of in-situ measurements, at center of Athens. However, most months have similar SSAs, with differences that lie well within the SSA variability of each month. Looking at the monthly mean AODs; despite the fact that standard deviations of both SSA and AOD's are large, it can be seen that higher AOD's are associated with less absorbing aerosol cases.

[Figure]

**Figure 9** Mean monthly SSAs (left axis) in the UV (UVMFR) at two wavelengths and AOD

at 368nm from the UVMFR (right axis) for the whole 5-year period, at Athens, errorbars represent one standard deviation of the mean..

When calculating diurnal patterns of the SSA at UV and visible wavelengths for the Athens area, we observed a mean diurnal pattern with a variability of the order of 0.02 to 0.05 and having highest absorption (lowest SSA's) ±2 hours around noon (figure 10). Similar behavior can also be seen from AERONET retrieved SSA's having higher values observed during the early morning and late evening. However, the SZA limitation of the AERONET retrieval methodology leads to lack of measurement points around noon. To investigate the uncertainty in relation to UVMFR retrievals, the diurnal pattern was calculated for different SSA

bins according to the analysis in the previous section. In general, the daily pattern is clear for each bin and is mirrored by the AERONET inversion retrievals. However, the statistical  one standard deviation bars are quite large. These bars describ the variability of the SSA's during each hourly bin, but also include retrieval uncertainty.  We have to note that since no restriction has been introduce for UVMFR SSA retrievals at low AOD's the uncertainty related with these data becomes larger as seen also in figure 7.

[Figure]

**Figure 10** Diurnal patterns of SSA derived from the UVMFR and CIMEL measurements.

Mean values per hour plotted with errorbars at one standard deviation. Local time in

Athens is UTC+2(winter) UTC+3(summer)

In order to investigate the possible dependence of SSA on AOD, figure 11 shows the synchronous UVMFR and CIMEL SSA retrievals plotted against AOD at 440nm. We found that in general, SSA decreases with a decrease in optical extinction, although lower AOD's are also linked to higher uncertainties of retrieved SSA. We believe that this behavior reflects seasonal changes in the average aerosol composition in Athens. Indeed, the annual cycle of

SSA is the same as the AOD annual cycle having a maximum in summer and a minimum in winter. Studies of the SSA annual variability for other cities such as Ispra, Italy and

Thessaloniki, Greece (Arola et al., 2005, Bais et al., 2005) revealed the same trend, with low

SSA values (high absorption) associated with low AOD and reminiscent of mostly wintertime cases. It has to be noted that due to low AOD, uncertainties associated with the data obtained from both retrieval techniques (AERONET and UVMFR), are quite high. For higher AOD

(>0.6), CIMEL retrievals show an almost constant value of the SSA ~0.92, while lower values have been retrieved at smaller

AODs   . Similar results were reported by Krotkov et al. (2005b) when analyzing measurements derived at at AERONET calibration site in Greenbelt,

Maryland USA.

[Figure]

**Figure 11** Dependence of the calculated SSA from AOD measurements

We performed an analysis of the differences of SSAs between the visible and the UV parts of the spectrum based on aerosol characteristics using synchronous CIMEL and UVMFR SSA

retrievals and an aerosol classification scheme described in detail in Mielonen et al. (2009).

There, a classification of AERONET data was used in order to derive 6 aerosol types based on the SSA measurement at 440nm and the AE that was derived in the 440-870 nm wavelength range. Mielonen et al. (2009) used a visualization of this characterization, by plotting AE versus SSA for individual sites, and compared their results with the CALIPSO

(Omar et al., 2005) aerosol classification scheme obtaining good agreement. In addition, the difference between SSA at 440 nm and 1020 nm (similar to the approach applied by Derimian et al. (2008)), was implemented to better distinguish fine absorbing aerosols from coarse ones.

The main idea was to fill this SSA versus AE aerosol type related "space" with the differences of $SSA_{440}$-$SSA_{368}$ (SSADIFF) to investigate a possible link between SSA wavelength dependence and aerosol type. In figure 12 using the Mielonen et al. (2009) aerosol typing approach, we plot SSADIFF for different classes (colored scale), and separate aerosol types by areas in the SSA/AE plot . In addition, actual points of $SSA_{440}$ retrieved by the CIMEL

instrument are shown in order to categorize Athens results according to the classification scheme.

[Figure]

**Figure 12** Daily average $SSA_{440}$ (CIMEL) versus $AE_{(440-870nm)}$ . Colors represent different bins

    of the spectral differences of $SSA_{440nm}$- $SSA_{368nm}$.

The results  in figure 12 show that a mixture of aerosol types characterizes the ARSS site in

Athens, with $SSA_{440}$ values spanning all 6 sub-spaces. Analyzing the wavelength dependence of the SSA, by defining SSADIFF as the difference $SSA_{440} - SSA_{368}$, there is evidence that high negative SSADIFF values (that means that the SSA at UV wavelengths is equal or relatively higher than $SSA_{440}$) tend to occur towards high AEs. For these cases (green color in figure 11) we observe high absorption cases with AE's around 1, which can be attributed in to polluted dust aerosol events. Also the majority of cases which comply with the condition AE < 0.7 are found with lower SSA at UV by at least 0.05 compared to $SSA_{440}$. More specifically, dust cases (mainly during spring) can be identified due to the proximity of Athens to the Saharan desert (Gerasopoulos, et al., 2010), explaining this behavior of absorbing aerosols at UV with low AE. Russell, et al., (2010) reported results obtained from diverse datasets showing SSA wavelength dependency from the IR down to visible wavelengths. In addition, Bergstrom et al. (2007) presented SSA spectra for dust-containing aerosols campaigns (PRIDE and ACE-Asia) including AERONET measurements at sites that are affected by dust such as Cape Verde, Bahrain (Persian Gulf) and the Solar Village (Saudi Arabia). Both studies concluded that the SSA spectra for AERONET locations, dominated by desert dust decrease with decreasing wavelength. In addition, Russel et al., (2010) reported that SSA spectra for AERONET locations dominated by urban-industrial and biomass-burning aerosols decrease with increasing wavelength in line with the results of Bergstrom et al. (2007). Figure 12⁠1 also shows that similar SSA values can be found for 440nm and 368nm and for fine aerosol cases (AE>1.4).

In order to understand the potential relative contributions of dust and brown carbon better, we applied the method of Schuster et al., (2⁠2016⁠) to the AERONET measurements in Athens. This method separates contributions from black carbon, organic carbon, hematite and goethite, using to the retrieved refractive index at all available wavelengths, even in complex mixtures. Figure 12 13 shows the fractions of total aerosol volume attributed to these components, as well as the volume fractions accordingly. It is evident, according to this approach, that both brown carbon and mineral dust are likely absorbing components involved in the aerosol mixture in Athens, and brown carbon playing the more dominant role. Brown carbon highly absorbs in UV wavelengths and hardly any above 0.7nm (Kirchstetter et all, 2004). BrC fraction is higher at in OCTOBEROctober, but it has very large concentrations at the period March-June, which partly explains low SSA values at figure 9.

[Figure]

[Figure]

**Figure 13.** Total volume (in the upper plot) and volume fraction (in the lower plot) of absorbing aerosol components, as inferred by the method of Schuster et al. 2016. The retrieval gives the fractions for fine and coarse mode separately and here the contributions are shown as mode-weighted median value.

The utility of the AE for aerosol  extinction is that its value depends primarily on the size of the particles, ranging from a value of 4 for very small particles (Rayleigh scattering) to around 0 for very large particles (such as cloud drops). Thus AE for atmospheric aerosol  mixtures varies between limits specified by particle size. Various studies (e.g. Bergström et al., 2007) have used the Ångström Absorption Exponent (AAE) for studying the aerosol absorption wavelength dependence for different aerosol types and mixing (which is calculated similarly with the Ångström Exponent, only using Absorption Optical depth [AOD*(1-SSA)] instead of AOD). As the absorption AOD is a relatively smooth decreasing function with wavelength, it can be approximated with a power law wavelength dependence via the AAE which is defined as the negative of the slope of the absorption on a log-log plot. Investigating the  temporal variability of $AAE_{(440-870)}$ and $AAE_{(332-440)}$ Measurements of $AAE_{(440-870)}$ are found to lie between 0.9 and 1.5 ($2\sigma$) in accordance with the results of Bergström et al., (2007). $AAE_{(332-440)}$ in the UV range is very different from that in the visible, with values ranging from 1.4 to 5 ($2\sigma$). A direct comparison reveals that for the aerosol composition features of Athens, the AAEs are usually up to 4 times higher in the UV range than in the visible. This is due to a combination of the enhanced absorption (lower SSA's) that has been found in the UV, together with higher AOD's in this band.

Finally, we have calculated mean CIMEL SSA values for all four retrieved wavelengths (440nm, 673nm, 870nm and 1020nm) for the whole period under study, and synchronous (5 minute SSA averaged around the CIMEL measurement time) UVMFR SSAs at UV (332nm and 368nm). The results are shown in figure 14 with errorbars at $1\sigma$. Datasets of SSA retrievals are separated in 3 cases accordingly: a) all points (CIMEL L1.5 and all synchronous UVMFR data), b) measurements retrieved with AOD>0.2 (reduced uncertainty),  c) SSA retrievals for $AE_{340-440}$<0.7, to identify dust events and d) cases with AE>1.2 to include the fine mode cases. While for all cases the calculated standard deviation is quite high ($\geq 0.05$), there is a systematic SSA decrease in the UV range, and mean differences of 0.07 and 0.02 have been found when comparing SSA at the visible range and SSA at 332nm and 368nm respectively. Dust cases in particular show a spectral decrease in SSA with decreasing wavelength from 1022nm (CIMEL) down to 332nm (UVMFR). Fine mode cases show smaller spectral dependence ($SSA_{440nm}-SSA_{332nm}$ <0.03).

[Figure]

**Figure 14** Wavelength dependence of SSA from synchronous CIMEL and UVMFR measurements. Blue points represent all data points, red data retrievals with AOD>0.2, green points data with AE>1.2 and black data only dust aerosol cases. Vertical bars represent one standard deviation of the calculated mean.

The spectral dependence of the SSA from the visible to the UV wavelengths is in agreement with findings presented by Corr et al., (2009) and Krotkov et al., (2009).. With Tthe same approach applied to Mexico City where measurements are also influenced by city emissions and blowing dust, Corr et al. (2009) studied the SSA behavior at UV wavelengths and showed that for AOD>0.1, SSA varied from 0.78 to 0.80 for 332nm and 368nm respectively with enhanced absorption at UV wavelengths relative to the visible wavelengths attributable to these types of aerosols. Krotkov et al., (2009) have modified a UVMFR in order to measure also at 440nm , and found strong SSA wavelength dependence across blue and near UV spectral region.

**4    Conclusions**

Advantages of measuring the aerosol absorption (SSA) in the UV with the UVMFR instrument can be summarized as follows:

- AOD, in the UV wavelength range, is higher (for the same aerosol mass) than in the visible spectral range

- SSA retrievals with the uncertainty of ±0.03 can be derived for SZA> 40 degrees and with an uncertainty of ±0.04 for all SZAs where AOD>=0.2

- SSA retrievals are stable and repeatable over the five year period

We have analyzed a 5 year period of UVMFR and CIMEL measurements at the city of Athens retrieving SSA at visible and UV wavelengths based on the effect of aerosol SSA on the Direct to Global Ratio (DGR) for a given AOD and air mass. Since the CIMEL retrieval algorithm is more accurate for high SZA, the combination of the two instruments allows for an increase in measurement frequency of SSA and the ability to derive a complete diurnal cycle of aerosol absorption. In addition, the spectral differences of the aerosol absorption properties in the visible and UV wavelength range have been investigated, using synchronous CIMEL and UVMFR retrievals. Results of this work confirmed similar results found for Mexico City, Mexico (Corr et al., 2009), Washington DC, USA (Krotkov et al., 2005b) and Rome, Italy (Ialongo et al., 2010), that presented enhanced absorption of aerosols for UV wavelengths.

We have also used the produced dataset to investigate possible effects of aerosol type on observed SSA wavelength differences. The enhanced UV absorption can be mainly due to either dust or organic aerosol. Our analysis of Athens AERONET measurements suggests that the relative role of absorbing organic aerosol would be somewhat more significant than dust. The enhanced aerosol absorption found when comparing UV and visible spectrum results, shows that:

- We expect a systematic overestimation of modeled solar UV irradiance using SSA from extrapolation from the visible range as an input to RTMs

- There is a possibility  of a decrease in specific days/cases of regional $O_3$ due to the enhanced aerosol absorption (Li et al., 2005). But for the Athens case this could be verified only with Chemical model results.

- Satellite post-correction  results (e.g. Arola et al., 2009), including aerosol absorption effects, have to take into account absorption enhancement in the UV range.

- We expect an overestimation on the UV irradiance (UV Index) calculations on cloudless cases under dust and/or brown carbon presence when using SSA values from the visible range. This as a combination of the overestimated SSA and the high AODs during such events.

However, the spectral SSA differences, that we found, are well within the uncertainty of both retrievals as instrumental effects or absolute calibration uncertainties of sky radiances (~5% for the CIMEL almucantar measurements) might also play an important role when performing such comparisons. The coincidence of AOD measurements, from both instruments, using a single ETC for various SZA over the extended 5 year period used here, is a sign that no systematic SZA dependent factors influence the final SSA results.

The extended SSA dataset significantly improves comparative statistics and provides additional information on the effect of varying background aerosol conditions and higher aerosol absorption than that provided by Washington, DC, where dust aerosol cases are very rare. In conclusion, the combined use of CIMEL sun and sky radiance measurements in the visible with UVMFR total and diffuse irradiance measurements in the UV, provide an important advantage for remote measurements of column aerosol absorption over the UV-Visible spectral range.

**Acknowledgements**

P.Raptis would like to acknowledge the project «Aristotelis- SOLAR (50561), Investgation on the factors affecting the solar radiation field in Greece». V. Amiridis and S. Kazadzis would like to acknowledge the project "European Union's Horizon 2020 Research and Innovation Programme ACTRIS-2 (grant agreement no. 654109)"